# Long-Tailed Partial Label Learning via Dynamic Rebalancing

**Feng Hong**[1]    **Jiangchao Yao**[1,2,†]    **Zhihan Zhou**[1]    **Ya Zhang**[1,2]    **Yanfeng Wang**[1,2,†]

[1]Cooperative Medianet Innovation Center, Shanghai Jiao Tong University
[2]Shanghai AI Laboratory

{feng.hong, Sunarker, zhihanzhou, ya_zhang, wangyanfeng}@sjtu.edu.cn

## Abstract

Real-world data usually couples the label ambiguity and heavy imbalance, challenging the algorithmic robustness of partial label learning (PLL) and long-tailed learning (LT). The straightforward combination of LT and PLL, *i.e.,* LT-PLL, suffers from a fundamental dilemma: LT methods build upon a given class distribution that is unavailable in PLL, and the performance of PLL is severely influenced in long-tailed context. We show that even with the auxiliary of an oracle class prior, the state-of-the-art methods underperform due to an adverse fact that the *constant rebalancing* in LT is harsh to the label disambiguation in PLL. To overcome this challenge, we thus propose a *dynamic rebalancing* method, termed as RECORDS, without assuming any prior knowledge about the class distribution. Based on a parametric decomposition of the biased output, our method constructs a dynamic adjustment that is benign to the label disambiguation process and theoretically converges to the oracle class prior. Extensive experiments on three benchmark datasets demonstrate the significant gain of RECORDS compared with a range of baselines. The code is publicly available.

## 1 Introduction

Partial label learning (PLL) origins from the real-world scenarios, where the annotation for each sample is an ambiguous set containing the groundtruth and other confusing labels. This is common when we gather annotations of samples from news websites with several tags (Luo & Orabona, 2010), videos with several characters of interest (Chen et al., 2018), or labels from multiple annotators (Gong et al., 2018). The ideal assumption behind PLL is that the collected data is approximately uniformly distributed regarding classes. However, a natural distribution assumption in above real-world applications should be imbalance, especially follows the long-tailed law, which should be considered if we deploy the PLL methods into online systems. This thereby poses a new challenge about the robustness of algorithms to both category imbalance and label ambiguity in PLL studies.

Existing efforts, partial label learning and long-tailed learning, independently study the partial aspect of this problem in the past decades. The standard PLL requires the label disambiguation from candidate sets along with the training of an ordinary classifier (Feng et al., 2020). The mainstream to solve this problem is estimating label-wise confidence to implicitly or explicitly re-weight the classification loss, *e.g.,* PRODEN (Lv et al., 2020), LW (Wen et al., 2021), CAVL (Fei et al., 2022) and CORR (Wu et al., 2022), which have achieved the state-of-the-art performance in PLL. When it comes to the long-tailed learning, the core difficulty lies on diminishing the inherent bias induced by the heavy class imbalance (Chawla et al., 2002; Menon et al., 2013). The simple but fairly effective method is the logit adjustment (Menon et al., 2021; Ren et al., 2020), which has been demonstrated very powerful in a range of recent studies (Cui et al., 2021; Narasimhan & Menon, 2021).

Nevertheless, considering a more practical long-tailed partial label learning (LT-PLL) problem, several dilemma remains based on the above two paradigms. One straightforward concern is that the skewed long-tailed distribution exacerbates the bias to the head classes in the label disambiguation,

---

[†] Corresponding to: Jiangchao Yao (Sunarker@sjtu.edu.cn) and Yanfeng Wang (wangyanfeng@sjtu.edu.cn).

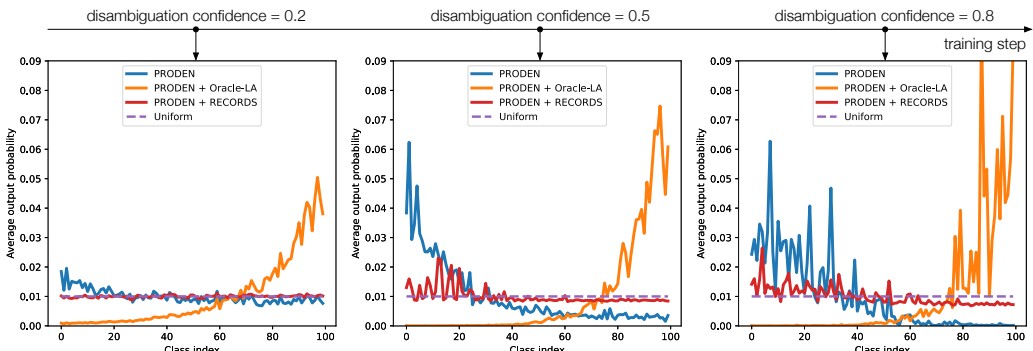

Figure 1: Average classifier prediction (on the CIFAR-100 test set) of different methods during training (on LT-PLL training set CIFAR-100-LT with imbalance ratio $\rho = 100$ and ambiguity $q = 0.05$). "PRODEN" (Lv et al., 2020) is a popular PLL method. "PRODEN + Oracle-LA" denotes PRODEN with the state-of-the-art logit adjustment (Menon et al., 2021; Hong et al., 2021) in LT under the oracle prior. "PRODEN + RECORDS" is PRODEN with our proposed calibration. "Uniform" characterizes the expected average confidence on different classes.

easily resulting in the trivial solution that are excessively confident to the head classes. More importantly, most state-of-the-art long-tailed learning methods cannot be directly used in LT-PLL, since they require the class distribution available that is agnostic in PLL due to the label ambiguities. In addition, we discover that even after applying an oracle class distribution prior in the training, existing techniques underperform in LT-PLL and even fail in some cases.

In Figure 1, we trace the average prediction of a PLL model PRODEN (Lv et al., 2020), on a uniform test set. Normally, the backbone PLL method PRODEN exhibits the biased prediction towards head classes shown by the blue curve, and ideally, we expect with the intervention of the state-of-the-art logit adjustment in LT, the prediction for all classes will be equally confident, namely, the purple curve. However, as can be seen, PRODEN calibrated by the oracle prior actually performs worse and is prone to over-adjusting towards the tail classes as shown in the orange curve. This is because logit adjustment in LT leverages a constant class distribution prior to rebalance the training and does not consider the dynamic of label disambiguation. Specially, at the early stage where the true label is very ambiguous from the candidate set, over-adjusting the logit only leads to the strong confusion of the classifier, which is negative to the overall training. Thus, we can see in Figure 1, the average prediction on the tail classes becomes too high along with the training.

Based on the above analysis, compared with the previous *constant rebalancing* methods in PLL, a *dynamic rebalancing* mechanism friendly to the training dynamic will be more preferred. To this intuition, we propose a novel method, termed as REbalanCing fOR Dynamic biaS (RECORDS) for LT-PLL. Specifically, we perform a parametric decomposition of the biased model output and implement a dynamic adjustment by maintaining a prototype feature with momentum updates during training. The empirical and theoretical analysis demonstrate that our dynamic parametric class distribution is asymmetrically approaching to the statistical prior but benign to the overall training. A quick glance at the performance of RECORDS is the red curve in Figure 1, which approximately fits the expected purple curve in the whole training progress.

The contribution can be summarized as follows,

1. We delve into a more practical but under-explored LT-PLL scenario, and identify its several challenges in this task that cannot be addressed and even lead to failure by the straightforward combination of the current long-tailed learning and partial label learning.

2. We propose a novel RECORDS for LT-PLL that conducts the dynamic adjustment to rebalance the training without requiring any prior about the class distribution. The theoretical and empirical analysis show that the dynamic parametric class distribution is asymmetrically approaching to the oracle class distribution but more friendly to label disambiguation.

3. Our method is orthogonal to existing PLL methods and can be easily plugged into the current PLL methods in an end-to-end manner. Extensive experiments on three benchmark datasets under the long-tailed setting and a range of PLL methods demonstrate the effectiveness of the proposed RECORDS. Specially, we show a $32.03\%$ improvement in classification performance compared to the best CORR on the Pascal VOC dataset.

## 2 RELATED WORK

**Partial Label Learning (PLL).** In PLL, each training sample is associated with a candidate label set containing the ground-truth. Early explorations is mainly average-based (Hüllermeier & Beringer, 2006; Zhang & Yu, 2015; Cour et al., 2011b), which treat all candidate labels equally during model training. Their drawback is that the model training is easily misled by the false positive labels that co-occur with the ground truth. Some other identification-based methods consider the truth label as a latent variable and optimize the objective under some criterions (Jin & Ghahramani, 2002; Liu & Dietterich, 2014; Nguyen & Caruana, 2008; Yu & Zhang, 2015). Recently, self-training methods (Feng et al., 2020; Wen et al., 2021; Lv et al., 2020; Fei et al., 2022) that gradually disambiguate the candidate label sets during training have achieved better results. PiCO (Wang et al., 2022b) introduces a contrastive learning branch that improves PLL performance by enhancing representation.

**Long-Tailed Learning (LT).** In LT, several methods have been proposed to consider the extremely skewed distribution during training (Cui et al., 2019; Zhou et al., 2022; Chawla et al., 2002). Re-sampling (Kubat & Matwin, 1997; Wallace et al., 2011; Han et al., 2005) is one of the most widely used paradigm by down-sampling samples of head classes or up-sampling samples of tail classes. Re-weighting (Morik et al., 1999; Menon et al., 2013) adjusts the sample weights in the loss function during training. Transfer learning (Chu et al., 2020; Wang et al., 2021; Kim et al., 2020) seeks to transfer knowledge from head classes to tail classes to obtain a more balanced performance. Recently, logit adjustment techniques (Menon et al., 2021; Ren et al., 2020; Tian et al., 2020) that modify the output logits of the model by an offset term $\log \mathbb{P}_{train}(y)$ have been the state-of-the-art.

**Long-Tailed Partial Label Learning (LT-PLL).** A few works approximately explore the LT-PLL problem. Liu et al. (2021) implicitly alleviates data imbalance in the non-deep PLL by constraining the parameter space, which is hard to apply in deep learning context due to the complex optimization. Concurrently, SoLar (Wang et al., 2022a) improves the label disambiguation process in LT-PLL through the optimal transport technique. However, it requires an extra outer-loop to refine the label prediction via sinkhorn-knopp iteration, which increases the algorithm complexity. Different from these works, this paper tries to solve the LT-PLL problem from the perspective of rebalancing.

## 3 PRELIMINARIES

### 3.1 PROBLEM FORMULATION

Let $\mathcal{X}$ be the input space and $\mathcal{Y} = \{1, 2, ..., C\}$ be the class space. The candidate label set space $\mathcal{S}$ is the powerset of $\mathcal{Y}$ without the empty set: $\mathcal{S} = 2^{\mathcal{Y}} - \emptyset$. A long-tailed partial label training set can be denoted as $\mathcal{D}_{train} = \{(\boldsymbol{x}_i, y_i, S_i)\}_{i=1}^N \in (\mathcal{X}, \mathcal{Y}, \mathcal{S})^N$, where any $\boldsymbol{x}_i$ is associated with a candidate label set $S_i \subset \mathcal{Y}$ and its ground truth $y_i \in S_i$ is invisible. The sample number $N_c$ of each class $c \in \mathcal{Y}$ in the descending order exhibits a long-tailed distribution. Let $f(\boldsymbol{x}; \theta)$ denote a deep model parameterized by $\theta$, transforming $x$ into an embedding vector. Then, the final output logits of $\boldsymbol{x}$ are given by $z(\boldsymbol{x}) = g(f(\boldsymbol{x}; \theta); \boldsymbol{W}) = \boldsymbol{W}^\top f(\boldsymbol{x}; \theta)$ where $g$ is a linear classifier with the parameter matrix $\boldsymbol{W}$. We leverage $\Theta = [\theta, \boldsymbol{W}]$ to denote all parameters of the deep network. For evaluation, a class-balanced test set $\mathcal{D}_{uni} = \{(\boldsymbol{x}_i, y_i)\}$ is used (Brodersen et al., 2010). In a nutshell, the goal of LT-PLL is to learn $\Theta$ on $\mathcal{D}_{train}$ that minimizes the following balanced error rate (BER) on $\mathcal{D}_{uni}$:

$$\min_{\Theta} \text{BER}(\Theta) \overset{\mathbb{P}_{uni}(y)=\frac{1}{C}}{=} \mathbb{P}_{(\boldsymbol{x},y)\in\mathcal{D}_{uni}}(y \neq \arg\max_{y'\in\mathcal{Y}} z^{y'}(\boldsymbol{x})). \tag{1}$$

### 3.2 SELF-TRAINING PLL METHODS

Self-training PLL methods maintain class-wise confidence weights $\boldsymbol{w}$ for each sample during training and formulate the loss function as a weighted summation of the classification loss by $\boldsymbol{w}$. $\boldsymbol{w}$ is updated in each training round based on the model output and gradually converges to the one-hot labels, converting PLL to ordinary classification. Specially, PRODEN (Lv et al., 2020) assigns higher confidence weights to labels with larger output logit; LW (Wen et al., 2021) additionally considers the loss of non-partial labels and assigns the corresponding confidence weights to non-partial labels as well; CAVL (Fei et al., 2022) designs a identification strategy based on the idea of CAM and assigns hard confidence weights to labels; and CORR (Wu et al., 2022) applies consistency regularization in the disambiguation strategy. Please refer to Appendix B.1 for more details.

### 3.3 LOGIT ADJUSTMENT FOR LONG-TAILED LEARNING

We briefly review logit adjustment (LA) (Menon et al., 2021; Hong et al., 2021), a powerful technique for supervised long-tailed learning. First, according to the Bayes rule, we have the underlying

class-probability $\mathbb{P}(y|\boldsymbol{x}) \propto \mathbb{P}(\boldsymbol{x}|y) \cdot \mathbb{P}(y)$. Directly minimizing the cross-entropy loss to reach the optimal has $\mathrm{softmax}(z^y(\boldsymbol{x})) = \mathbb{P}_{train}(y|\boldsymbol{x})$ (Yu et al., 2018). However, for a BER-optimal output logit $z_{uni}$, it is about the balanced data, which has $\mathrm{softmax}(z_{uni}^y(\boldsymbol{x})) = \mathbb{P}_{uni}(y|\boldsymbol{x}) \propto \mathbb{P}(\boldsymbol{x}|y) \cdot \mathbb{P}_{uni}(y)$ and $\mathbb{P}_{uni}(y) = \frac{1}{C}$ (Menon et al., 2013). Then, we have the following relations

$$
\begin{aligned}
\mathbb{P}_{uni}(y|\boldsymbol{x}) &\propto \mathbb{P}(\boldsymbol{x}|y) \cdot \mathbb{P}_{train}(y) \,/\, \mathbb{P}_{train}(y) \\
&\propto \mathbb{P}_{train}(y|\boldsymbol{x}) \,/\, \mathbb{P}_{train}(y) \\
&\propto \mathrm{softmax}(z^y(\boldsymbol{x}) - \log \mathbb{P}_{train}(y)).
\end{aligned}
\tag{2}
$$

That is to say, if we have the logit $z^y(\boldsymbol{x})$ by training on standard cross-entropy loss, a BER-optimal logit $z_{uni}^y(\boldsymbol{x}) = z^y(\boldsymbol{x}) - \log \mathbb{P}_{train}(y)$ can be obtained by subtracting an offset term $\log \mathbb{P}_{train}(y)$. Using $z_{uni}^y(\boldsymbol{x})$ as the test logit preserves the balanced part $\mathbb{P}(\boldsymbol{x}|y)$ of the output and removes the imbalanced part $\mathbb{P}_{train}(y)$ in a statistical sense. LA has demonstrated its effectiveness on a range of recent long-tailed learning methods (Zhu et al., 2022; Cui et al., 2021).

## 4 METHOD

In LT-PLL, the model output is biased to head classes, and using biased output to update confidence weights will result in biased re-weighting, namely, a tendency to identify a more frequent label in the candidate set as the ground truth. In turn, biased re-weighting on the loss function will further lead to a more severe imbalance on the model. In this section, we seek to propose a universal rebalancing algorithm that obtains unbiased output $z_{uni}$ from biased output $z$ in self-training PLL methods.

### 4.1 MOTIVATION

Previous long-tailed learning techniques assume no ambiguity in supervision and focus only on the model bias caused by the training set distribution, which we term as *constant rebalancing*. As shown in Figure 1, the constant rebalancing (Oracle-LA) fails to rebalance model in LT-PLL. This is because in LT-PLL, not only the skewed training set distribution, but also the ambiguous supervision can affect the training, since the inferred label changes continuously along with the label disambiguation process. Specially, at the early stage, the prediction imbalance of PRODEN is not significant. Using the more skewed training set distribution instead leads to a model that is heavily biased towards the tail classes, inducing the difficulty in label disambiguation. Therefore, a *dynamic rebalancing* method that considers the label disambiguation process can be intuitively more effective, *e.g.,* RECORDS in Figure 1. In the following, we will concretely present our method.

### 4.2 RECORDS: REBALANCING FOR DYNAMIC BIAS

The above analysis empirically suggests that the constant calibration by LA cannot match the dynamic of label disambiguation in LT-PLL. Formally, Equation 2 fails because the label disambiguation dynamically affects model optimization during training, which induces the mismatch between the ground truth $\mathbb{P}_{train}(y)$ and the training dynamics of the label disambiguation. To mediate the conflict, we propose a parametric decomposition on the original rebalancing paradigm:

$$
\begin{aligned}
\mathbb{P}_{uni}(y|\boldsymbol{x}; \Theta) &\propto \mathbb{P}(\boldsymbol{x}|y; \Theta) \cdot \mathbb{P}_{train}(y|\Theta) \,/\, \mathbb{P}_{train}(y|\Theta) \\
&\propto \mathbb{P}_{train}(y|\boldsymbol{x}; \Theta) \,/\, \mathbb{P}_{train}(y|\Theta) \\
&\propto \mathrm{softmax}(z^y(\boldsymbol{x}) - \log \mathbb{P}_{train}(y|\Theta)),
\end{aligned}
\tag{3}
$$

where $\mathbb{P}_{uni}(y|\boldsymbol{x}; \Theta)$ is the parametric class-probabilities under a uniform class prior. Here, we start a perspective of dynamic rebalancing: a dynamic $\mathbb{P}_{train}(y|\Theta)$ adapted to the training process is required instead of the constant prior $\mathbb{P}_{train}(y)$ to rebalance the model in LT-PLL. Existing PLL methods work to achieve better disambiguation results or improve the quality of the representation, *i.e.,* learning $\mathbb{P}(\boldsymbol{x}|y; \Theta)$ that is closer to $\mathbb{P}(\boldsymbol{x}|y)$. Thus, our study is orthogonal to existing PLL methods and can be combined together to improve the performance (see Section 5.2). Also, the rebalanced output can boost the accuracy of label disambiguation and a better $\mathbb{P}(\boldsymbol{x}|y; \Theta)$ can be learned.

Here, our effective and lightweight design is the estimation of the dynamic class distribution $\mathbb{P}_{train}(y|\Theta)$ by the model via the training set, i.e., $\mathbb{P}_{train}(y|\Theta) = \mathbb{E}_{\boldsymbol{x}_i \in \mathcal{D}_{train}} \mathbb{P}_{train}(y|\boldsymbol{x}_i; \Theta)$. First,

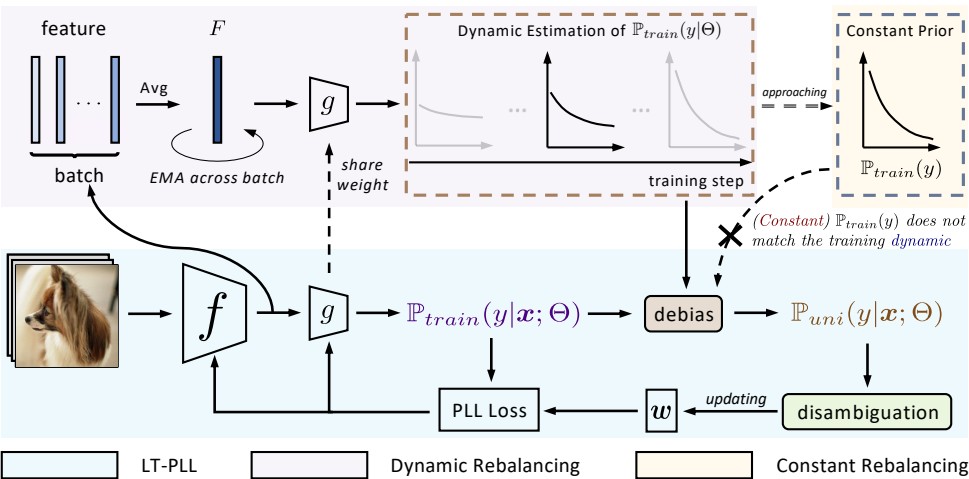

Figure 2: Illustration for RECORDS. The class-wise confidence weights $\boldsymbol{w}$ are updating by the "disambiguation" module for each sample and used as soft/hard pseudo labels in PLL Loss. The main differences between the PLL baselines are the "PLL Loss" and the "disambiguation" module (see Table 6). The "debias" module dynamically rebalances $\mathbb{P}_{train}(y|\boldsymbol{x};\Theta)$ to obtain $\mathbb{P}_{uni}(y|\boldsymbol{x};\Theta)$. A balanced $\mathbb{P}_{uni}(y|\boldsymbol{x};\Theta)$ helps tail samples to disambiguate labels more accurately and avoid being overwhelmed by head classes. A momentum-updated prototype feature is used to estimate $\mathbb{P}_{train}(y|\Theta)$, which is benign to label disambiguation and asymmetrically approaching the oracle prior $\mathbb{P}_{train}(y)$. In comparison, constant rebalancing does not consider the dynamic of the label disambiguation.

we use the Normalized Weighted Geometric Mean (NWGM) approximation (Baldi & Sadowski, 2013) to put the expectation operation inside the softmax as follows:

$$\mathbb{P}_{train}(y|\Theta) = \mathbb{E}_{\boldsymbol{x}_i \in \mathcal{D}_{train}} \text{softmax}(z^y(\boldsymbol{x}_i)) \overset{NWGM}{\approx} \text{softmax}(\mathbb{E}_{\boldsymbol{x}_i \in \mathcal{D}_{train}} z^y(\boldsymbol{x}_i))$$
$$= \text{softmax}(g^y(\mathbb{E}_{\boldsymbol{x}_i \in \mathcal{D}_{train}} f(\boldsymbol{x}_i;\theta); \boldsymbol{W})). \quad (4)$$

Note that, NWGM approximation is widely used in dropout understanding (Baldi & Sadowski, 2014), caption generation (Xu et al., 2015), and causal inference (Wang et al., 2020). The intuition behind Equation 4 is to capture more stable feature statistics by means of NWGM and combine with the latest updated linear classifier to estimate $\mathbb{P}_{train}(y|\Theta)$. Nevertheless, with Equation 4, we do not reach the final form, since directly estimating $\mathbb{P}_{train}(y|\Theta)$ requires to consider the whole dataset via an EM alternation. To improve the efficiency, we design a momentum mechanism to accumulatively compute the expectation of features along with the training. Concretely, we maintain a prototype feature $F$ for the entire training set, using each batch's feature expectation for momentum updates:

$$F \leftarrow mF + (1-m)\mathbb{E}_{\boldsymbol{x}_i \in Batch} f(\boldsymbol{x}_i;\theta), \quad (5)$$

where $m \in [0,1)$ is a momentum coefficient. Then, replacing $\mathbb{E}_{\boldsymbol{x}_i \in \mathcal{D}_{train}} f(\boldsymbol{x}_i;\theta)$ in Equation 4 by $F$ yields the final implementation of our method:

$$z^y_{uni}(\boldsymbol{x}) = z^y(\boldsymbol{x}) - \log \mathbb{P}_{train}(y|\Theta) = z^y(\boldsymbol{x}) - \log \text{softmax}(g^y(F; \boldsymbol{W})). \quad (6)$$

Our RECORDS is lightweight and can be easily plugged into existing PLL methods in an end-to-end manner. As illustrated in Figure 2, we insert a "debias" module before label disambiguation of each training iteration, converting $\mathbb{P}_{train}(y|\boldsymbol{x};\Theta)$ to $\mathbb{P}_{uni}(y|\boldsymbol{x};\Theta)$ via Equation 6.

### 4.3 RELATION BETWEEN DYNAMIC REBALANCING AND CONSTANT REBALANCING

In previous sections, we design a parametric class distribution to calibrate training in LT-PLL. However, it is not clear about the relation of $\mathbb{P}_{train}(y|\Theta)$ and $\mathbb{P}_{train}(y)$. In this section, we theoretically point out their connection. First, let $\mathbb{P}_{train}(y_j) = \mathbb{E}_{(\boldsymbol{x},y)\sim(\mathcal{X},\mathcal{Y})}\mathbb{1}(y=y_j), y_j \in \mathcal{Y}$ denote the oracle prior, where $\mathbb{1}(\cdot)$ denotes the indicator function. Considering a hypothesis space $H$ where each $h_\Theta \in H : \mathcal{X} \to \mathcal{Y}$ is a multiclass classifier parameterized by $\Theta$ ($h = g \circ f$ in Figure 2), we define the parametric class distribution for $h_\Theta$ as $\mathbb{P}_{train}(y_j|\Theta) = \mathbb{E}_{(\boldsymbol{x},y)\sim(\mathcal{X},\mathcal{Y})}\mathbb{1}(h_\Theta(\boldsymbol{x})=y_j), y_j \in \mathcal{Y}$. Assume the disambiguated label for $(\boldsymbol{x}, y, S)$ in LT-PLL is $\tilde{y}(\boldsymbol{x}, S) \in S$ during training. Then, the empirical risk on basis of the disambiguated label under $\mathcal{D}_{train}$ is $R^{\mathcal{D}_{train}}(\Theta) = \frac{1}{N}\sum_{i=1}^{N} \mathbb{1}(h_\Theta(\boldsymbol{x}_i) \neq$

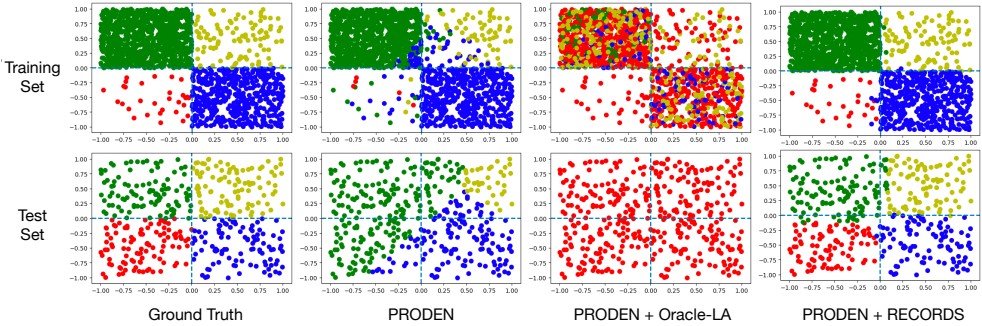

Figure 4: Visualization of the toy study. The first column illustrates the groud truth distribution of four classes on the training set and on the test set, marked by color. The right three columns exhibit the label identification from the candidate set for training samples in the first row and the predictions on the test set in the second row *w.r.t.* three methods.

$\tilde{y}(\boldsymbol{x}_i, S_i))$. Motivated by the theoretical study on the learnability of PLL (Liu & Dietterich, 2014), we propose Proposition 1 to discuss the relation between dynamic and constant rebalancing.

**Proposition 1.** *Let* $\eta = \sup_{(\boldsymbol{x},y)\in\mathcal{X}\times\mathcal{Y}, y_j\in\mathcal{Y}, y_j\neq y} \mathbb{P}_{S|(\boldsymbol{x},y)}(y_j \in S)$ *denote the ambiguity degree,* $d_H$ *be the Natarajan dimension of the hypothesis space $H$, $\tilde{h} = h_{\tilde{\Theta}}$ be the optimal classifier on the basis of the label disambiguation, where $\tilde{\Theta} = \arg\min_{\Theta} R^{\mathcal{D}_{train}}(\Theta)$. If the small ambiguity degree condition (Cour et al., 2011a; Liu & Dietterich, 2014)) satisfies, namely, $\eta \in [0,1)$, then for $\forall \delta > 0$, the $L_2$ distance between $\mathbb{P}_{train}(y)$ and $\mathbb{P}_{train}(y|\tilde{\Theta})$ given $\tilde{h}$ is bounded as*

$$L_2\left(\tilde{h}\right) < \frac{4}{(\ln 2 - \ln(1+\eta))N}(d_H(\ln 2N + 2\ln C) - \ln\delta + \ln 2)$$

*with probability at least $1 - \delta$, where $N$ is the sample number and $C$ is the category number.*

Proposition 1 yields an important implication that alongside the label disambiguation, the dynamic estimation can progressively approach to the oracle class distribution under small ambiguity degree. We kindly refer the readers to Appendix D for the complete proof. To further verify this, we trace the $L_2$ distance between the estimation of $\mathbb{P}_{train}(y|\Theta)$ and the oracle $\mathbb{P}_{train}(y)$ during training in each epoch and visualize it in Figure 3(a). It can be observed that the distance between two distributions

are gradually minimized, *i.e.,*, our parametric class distribution gradually converges to the statistical oracle class prior. Besides, as shown in Figure 3(b), the final estimated class distribution is very close to the oracle class prior. In total, Figure 3 indicates that our dynamic rebalancing is not only benign to the label disambiguation at the early stages, but also finally approach to the constant rebalancing.

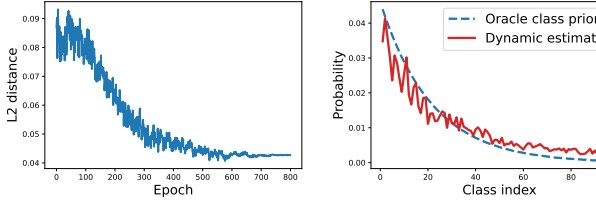

(a) $L_2$ distance during training    (b) Estimated class distribution

Figure 3: (a) $L_2$ distance between the estimated class distribution and the oracle class prior during training. (b) The final estimated class distribution. Experiment is conducted by CORR + RECORDS on CIFAR-100-LT with imbalance ratio $\rho = 100$ and ambiguity $q = 0.03$.

## 5 EXPERIMENTS

### 5.1 TOY STUDY

We simulate a four-class LT-PLL task where each class is distributed in different regions and their samples distribute uniformly in the corresponding regions. *The label set of each sample* consists of a true label and negative labels having a 0.6 probability flipped from other classes. In the first column of Figure 4, we visualize the training set and the test set: the training set is highly imbalanced among classes, with the number of samples from each class being 30 (red), 100 (yellow), 500 (blue) and 1000 (green), while the test set is balanced. In the right three columns of Figure 4, we show the results of identifying true labels from the candidate sets on the training set and the prediction results on the test set for different methods respectively. As can be seen, PRODEN results in the minority category (red) being completely overwhelmed by the majority categories (blue and green), and the

Table 1: Top-1 accuracy on three benchmark datasets. Bold indicates the superior results.

| | CIFAR-10-LT | | | | | | CIFAR-100-LT | | | | | | PASCAL VOC |
|---|---|---|---|---|---|---|---|---|---|---|---|---|---|
| Imbalance ratio $\rho$ | 50 | | | 100 | | | 50 | | | 100 | | | |
| Ambiguity $q$ | 0.3 | 0.5 | 0.7 | 0.3 | 0.5 | 0.7 | 0.03 | 0.05 | 0.07 | 0.03 | 0.05 | 0.07 | |
| CORR | 76.12 | 56.45 | 41.56 | 66.38 | 50.09 | 38.11 | 42.29 | 38.03 | 36.59 | 38.39 | 34.09 | 31.05 | 24.43 |
| + Oracle-LA post-hoc | 80.70 | 58.49 | 43.44 | 72.96 | 54.64 | 41.66 | 46.94 | 40.76 | 39.07 | 41.49 | 36.79 | 33.32 | 34.12 |
| + Oracle-LA | 36.27 | 17.61 | 12.77 | 29.97 | 15.80 | 11.75 | 22.56 | 5.59 | 3.12 | 11.37 | 3.32 | 1.98 | 52.51 |
| + RECORDS | **82.57** | **80.28** | **67.24** | **77.66** | **72.90** | **57.46** | **48.06** | **45.56** | **42.51** | **42.25** | **40.59** | **38.65** | **56.46** |
| vs. CORR | +6.45 | +23.83 | +25.68 | +11.28 | +22.81 | +19.35 | +5.77 | +7.53 | +5.92 | +3.86 | +6.40 | +7.60 | +32.03 |
| PRODEN | 73.12 | 54.45 | 41.37 | 63.55 | 47.37 | 38.06 | 39.23 | 35.45 | 33.90 | 34.52 | 32.04 | 29.40 | 22.39 |
| + Oracle-LA post-hoc | 77.41 | 57.14 | 42.91 | 70.71 | 48.79 | 41.38 | 43.40 | 38.64 | 35.82 | 38.40 | 35.20 | 31.92 | 31.53 |
| + Oracle-LA | 27.18 | 16.97 | 11.52 | 19.51 | 14.11 | 11.17 | 12.37 | 4.09 | 2.64 | 6.79 | 2.73 | 1.98 | 48.33 |
| + RECORDS | **79.48** | **76.73** | **65.31** | **72.15** | **65.22** | **52.26** | **44.56** | **41.31** | **39.26** | **39.13** | **37.23** | **35.26** | **52.65** |
| vs. PRODEN | +6.36 | +22.28 | +23.94 | +8.60 | +17.85 | +14.2 | +5.33 | +5.86 | +5.36 | +4.61 | +5.19 | +5.86 | +30.26 |
| LW | 70.11 | 37.67 | 22.73 | 64.78 | 39.57 | 23.54 | 35.54 | 29.50 | 27.86 | 31.58 | 28.09 | 24.65 | 19.41 |
| + Oracle-LA post-hoc | 74.34 | 40.27 | 25.34 | 69.60 | 42.34 | 27.35 | 35.47 | 28.80 | 27.27 | 31.03 | 26.96 | 23.20 | 21.06 |
| + Oracle-LA | 41.90 | 21.36 | 15.28 | 25.75 | 20.35 | 14.24 | 30.37 | 14.43 | 4.79 | 30.30 | 5.08 | 2.70 | 51.53 |
| + RECORDS | **76.02** | **57.39** | **40.28** | **71.18** | **57.23** | **41.24** | **36.56** | **31.67** | **29.39** | **33.00** | **28.85** | **25.64** | **53.09** |
| vs. LW | +5.91 | +19.72 | +17.55 | +6.40 | +17.66 | +17.70 | +1.02 | +2.17 | +1.53 | +1.42 | +0.76 | +0.99 | +33.68 |
| CAVL | 56.73 | 40.27 | 18.52 | 54.28 | 38.97 | 17.28 | 29.63 | 17.31 | 8.34 | 28.29 | 25.39 | 8.20 | 17.25 |
| + Oracle-LA post-hoc | 55.23 | 39.76 | 18.34 | 51.37 | 37.28 | 14.58 | 29.65 | 14.86 | 5.76 | 28.34 | 26.27 | 5.80 | 22.27 |
| + Oracle-LA | 22.16 | 14.97 | 11.50 | 18.29 | 14.23 | 10.67 | 17.31 | 4.36 | 2.83 | 7.24 | 2.55 | 2.03 | 50.78 |
| + RECORDS | **67.27** | **61.23** | **40.71** | **64.35** | **58.27** | **37.38** | **42.25** | **36.53** | **29.13** | **36.93** | **31.49** | **24.98** | **53.07** |
| vs. CAVL | +10.54 | +20.96 | +22.19 | +10.07 | +19.30 | +20.1 | +12.62 | +19.22 | +14.27 | +8.64 | +6.10 | +16.78 | +35.82 |

Table 2: Fine-grained analysis on CIFAR-100-LT with $\rho = 100$ and $q \in \{0.03, 0.05, 0.07\}$. Many/Medium/Few corresponds to three partitions on the long-tailed data.

| Method | $q = 0.03$ | | | | $q = 0.05$ | | | | $q = 0.07$ | | | |
|---|---|---|---|---|---|---|---|---|---|---|---|---|
| | Many | Medium | Few | Overall | Many | Medium | Few | Overall | Many | Medium | Few | Overall |
| CORR | 68.43 | 37.40 | 4.50 | 38.39 | 67.51 | 29.60 | 0.33 | 34.09 | 68.86 | 19.80 | 0.07 | 31.05 |
| + Oracle-LA post-hoc | 70.37 | 41.89 | 7.33 | 41.49 | 70.46 | 33.40 | 1.47 | 36.79 | 69.77 | 24.86 | 0.67 | 33.32 |
| + Oracle-LA | 11.03 | 12.34 | 10.63 | 11.37 | 0.34 | 4.46 | 5.47 | 3.32 | 0.00 | 0.71 | 5.77 | 1.98 |
| + RECORDS | 66.37 | 42.54 | 13.77 | **42.25** | 68.49 | 40.20 | 8.50 | **40.59** | 69.97 | 36.71 | 4.37 | **38.65** |
| vs. CORR | -2.06 | +5.14 | +9.27 | +3.86 | +0.98 | +10.60 | +8.17 | +6.50 | +1.11 | +16.91 | +4.30 | +7.60 |

yellow category being dominated mostly. After calibration with the oracle class prior, PRODEN + Oracle-LA instead predicts all data in the test set as the minority class (red). This suggests that constant rebalancing is prone to being over-adjusted towards tail classes coupling with the label disambiguation dynamic. In comparison, when using RECORDS as a dynamic rebalancing mechanism in LT-PLL, we achieve the desired results on both the training set and the test set.

## 5.2 BENCHMARK RESULTS

**Long-tailed partial label datasets.** We evaluate RECORDS on three datasets: CIFAR-10-LT (Liu et al., 2019), CIFAR-100-LT (Liu et al., 2019) and PASCAL VOC. For CIFAR-10-LT and CIFAR-100-LT, we build the long-tailed version of of CIFAR-10/100 with imbalanced ratio $\rho \in \{50, 100\}$. Following Lv et al. (2020); Wen et al. (2021), we adopt the *uniform* setting in CIFAR-10-LT and CIFAR-100-LT, *i.e.*, $\mathbb{P}(\overline{y} \in S | \overline{y} \neq y) = q$ to generate candidate label set of each sample. PASCAL VOC is a real-world LT-PLL dataset constructed from PASCAL VOC 2007 (Everingham et al.). Specifically, we crop objects in images as instances and all objects appearing in the same original image are regarded as the labels of a candidate set and empirically we can observe the significant class imbalance. Note that, here we are the first to conduct experiments based on deep models on real-world datasets, while previous PLL real-world datasets are tabular and only suitable for linear models or shallow MLPs (Fei et al., 2022). Please refer to Appendix E.1 for more details.

**Baselines.** We consider the state-of-the-art PLL algorithms including PRODEN (Lv et al., 2020), LW (Wen et al., 2021), CAVL (Fei et al., 2022), and CORR (Wu et al., 2022) and their combination with the state-of-the-art rebalancing method logit adjustment (LA) (Menon et al., 2021; Hong et al., 2021). We use two rebalancing variants: (1) Oracle-LA: rebalance a PLL model during training with the oracle class prior by LA; (2) Oracle-LA post-hoc: rebalance a pre-trained PLL model with the oracle prior using LA in a post-hoc way. Note that, for our methods, we directly apply RECORDS into PLL baselines but without requiring the oracle class prior. For comparisons with concurrent LT-PLL work *SoLar* (Wang et al., 2022a), please refer to Appendix E.7.

**Implementation details.** We use 18-layer ResNet as the backbone. The standard data augmentations are applied as in Cubuk et al. (2020). The mini-batch size is set to 256 and all the methods are trained using SGD with momentum of 0.9 and weight decay of 0.001 as the optimizer. The hyper-parameter $m$ in Equation 5 is set to 0.9 constantly. The initial learning rate is set to 0.01. We train the model for 800 epochs with the cosine learning rate scheduling.

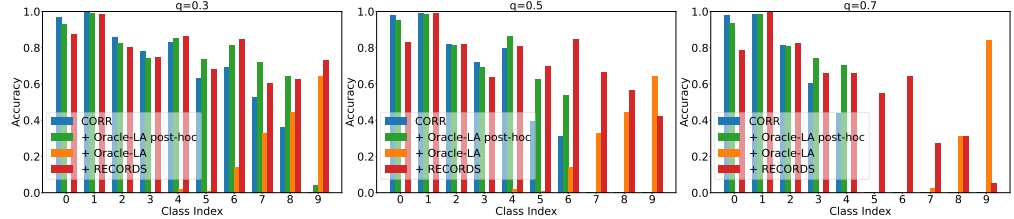

Figure 5: The per-class accuracy on CIFAR-10-LT with $\rho = 100$ and $q \in \{0.3, 0.5, 0.7\}$.

Table 3: Fine-grained analysis on CIFAR-100-LT-NU with $\rho = 100$ and $q \in \{0.03, 0.05, 0.07\}$.

| Method | $q = 0.03$ | | | | $q = 0.05$ | | | | $q = 0.07$ | | | |
|---|---|---|---|---|---|---|---|---|---|---|---|---|
| | Many | Medium | Few | Overall | Many | Medium | Few | Overall | Many | Medium | Few | Overall |
| CORR | 66.71 | 34.57 | 3.13 | 36.39 | 70.23 | 21.57 | 0.13 | 32.17 | 65.37 | 16.11 | 0.13 | 28.56 |
| + Oracle-LA post-hoc | 69.80 | 40.43 | 4.57 | 39.95 | 71.20 | 28.77 | 1.30 | 35.38 | 68.71 | 20.17 | 1.10 | 31.44 |
| + Oracle-LA | 3.26 | 9.97 | 13.10 | 8.56 | 0.86 | 5.34 | 12.27 | 5.85 | 0.00 | 0.25 | 16.87 | 5.92 |
| + RECORDS | 65.60 | 41.26 | 11.77 | **40.93** | 66.40 | 38.71 | 8.57 | **39.36** | 62.09 | 37.40 | 7.47 | **37.06** |
| vs. CORR | -1.11 | +6.69 | +8.44 | +4.54 | -3.83 | +17.14 | +8.44 | +7.19 | -3.28 | +21.29 | +7.34 | +8.50 |

Table 4: Comparision with more baselines for imbalanced learning.

| | CIFAR-10-LT | | | | | | CIFAR-100-LT | | | | | |
|---|---|---|---|---|---|---|---|---|---|---|---|---|
| Imbalance ratio $\rho$ | 50 | | | 100 | | | 50 | | | 100 | | |
| ambiguity $q$ | 0.3 | 0.5 | 0.7 | 0.3 | 0.5 | 0.7 | 0.03 | 0.05 | 0.07 | 0.03 | 0.05 | 0.07 |
| CORR | 76.12 | 56.45 | 41.56 | 66.38 | 50.09 | 38.11 | 42.29 | 38.03 | 36.59 | 38.39 | 34.09 | 31.05 |
| + Oracle-BCL | 43.76 | 25.24 | 17.34 | 33.23 | 18.34 | 15.72 | 27.23 | 10.34 | 8.34 | 14.34 | 5.82 | 4.10 |
| + Oracle-PaCO | 42.45 | 26.34 | 16.23 | 34.21 | 18.23 | 14.23 | 27.37 | 11.23 | 7.23 | 15.82 | 5.62 | 4.72 |
| + LDD | 75.23 | 58.02 | 42.14 | 67.59 | 51.38 | 39.28 | 42.91 | 39.05 | 36.98 | 39.20 | 34.87 | 32.15 |
| + SFN | 75.98 | 57.13 | 41.77 | 66.91 | 50.23 | 38.71 | 43.14 | 38.62 | 37.03 | 39.31 | 33.91 | 31.67 |
| + RECORDS | **82.57** | **80.28** | **67.24** | **77.66** | **72.90** | **57.46** | **48.06** | **45.56** | **42.51** | **42.25** | **40.59** | **38.65** |

**Overall performance.** In Table 1, we summarize the Top-1 accuracy on three benchmark LT-PLL datasets. Our method clearly exhibits superior performance on all datasets with different experimental settings under CORR, PRODEN, LW and CAVL. Specially, compared to the best PLL baseline CORR, RECORDS significantly improve the accuracy by 6.45%-25.68% on CIFAR-10-LT, 3.86%-7.60% on CIFAR-100-LT, and 32.03% on PASCAL VOC. When Oracle-LA is applied into PLL baselines, it induces severe performance degradation on CIFAR but improves significantly on PASCAL VOC. In comparison, Oracle-LA post-hoc can better alleviates the class imbalance on CIFAR. However, both of them cannot address the adverse effect of constant rebalancing on label disambiguation. In comparison, our RECORDS that solves this problem through dynamic rebalancing during training, achieves the consistent and the best improvements among all methods.

**Fine-grained analysis.** In Figure 5, we visualize the per-class accuracy on CIFAR-10-LT with $\rho = 100$ under the best PLL baseline CORR. As expected, dominant classes generally exhibits a higher accuracy in CORR. Oracle-LA post-hoc can improve CORR performance on medium classes, but accuracy remains poor on tail classes, especially when label ambiguity is high. Oracle-LA performs very poorly on head and medium classes due to over-adjustment towards tail classes. Our RECORDS systematically improves performance over CORR, particularly on rare classes. In Table 2, we show the Many-Medium-Few[1] accuracies of different methods on CIFAR-100-LT with $\rho = 100$. Our RECORDS shows the significant and consistent gains on Medium and Few classes when combined with CORR. As demonstrated in many long-tailed learning literatures (Kang et al., 2020; Menon et al., 2021), there might be a head-to-tail accuracy tradeoff in LT-PLL, and our method achieve the best overall accuracy in this tradeoff. More results are summarized in Appendix E.4.

### 5.3 FURTHER ANALYSIS

**Non-uniform candidates generation.** Real-world annotation ambiguity often occurs among semantically close labels. To evaluate the performance of RECORDS in practical scenarios, we conduct experiments on a more challenging dataset, namely CIFAR-100-LT-NU. To build CIFAR-100-LT-NU, we generate candidates from the ground truth of CIFAR-100-LT using a *Non-uniform* setting. Specifically, labels in the same superclass of the ground truth have a higher probability to be selected into the candidate set, *i.e.,* $\mathbb{P}(\overline{y} \in S | \overline{y} \neq y, D(\overline{y}) \neq D(y)) = q, \mathbb{P}(\overline{y} \in S | \overline{y} \neq y, D(\overline{y}) = D(y)) = 8q$, where $D(y)$ denotes the superclass to which $y$ belongs. In Table 3, we show the fine-grained analysis on CIFAR-100-LT-NU with $\rho = 100$ and $q \in \{0.03, 0.05, 0.07\}$. Combined with

---

[1]Many/Medium/Few corresponds to classes with $>100$, $[20, 100]$, and $<20$ images (Kang et al., 2020).

Table 5: Comparison with other dynamic strategies on CIFAR-10-LT and CIFAR-100-LT.

| | CIFAR-10-LT | | | | | | CIFAR-100-LT | | | | | |
|---|---|---|---|---|---|---|---|---|---|---|---|---|
| Imbalance ratio $\rho$ | 50 | | | 100 | | | 50 | | | 100 | | |
| Ambiguity $q$ | 0.3 | 0.5 | 0.7 | 0.3 | 0.5 | 0.7 | 0.03 | 0.05 | 0.07 | 0.03 | 0.05 | 0.07 |
| CORR | 76.12 | 56.45 | 41.56 | 66.38 | 50.09 | 38.11 | 42.29 | 38.03 | 36.59 | 38.39 | 34.09 | 31.05 |
| + Temp Oracle-LA | 81.37 | 43.62 | 18.10 | 76.09 | 25.88 | 16.11 | 47.44 | 43.46 | 29.75 | 41.78 | 39.19 | 33.69 |
| + Epoch RECORDS | 75.43 | 70.27 | 59.50 | 69.38 | 63.12 | 47.85 | 46.54 | 43.07 | 38.28 | 41.58 | 37.14 | 34.38 |
| + RECORDS | **82.57** | **80.28** | **67.24** | **77.66** | **72.90** | **57.46** | **48.06** | **45.56** | **42.51** | **42.25** | **40.59** | **38.65** |

(a) Linear Probing

(b) Ablation on $m$

Figure 6: (a) Top-1 accuracy (left) and top-5 accuracy (right) under different shots of linear probing for different methods pretrained on CIFAR-100-LT ($\rho = 100$, $q = 0.05$). CORR + Oracle-LA posthoc is same to PRODEN in terms of features and thus is not plotted. (b) Performance of CORR + RECORDS with varying $m$ on CIFAR-100-LT ($\rho = 100$, $q = 0.05$).

our RECORDS, the best baseline CORR achieves significant gains, demonstrating the robustness of RECORDS in different scenarios. More results are summarized in Appendix E.3.

**Linear probing performance.** Following the literatures of self-supervised learning (Chen et al., 2020; He et al., 2020), we conduct linear probing on CIFAR-100-LT with $\rho = 100$ and $q = 0.05$ to quantitatively evaluate the representation quantity of different methods. To eliminate the class imbalance effect, the linear classifier is trained on a balanced dataset. From Figure 6(a), our RECORDS consistently outperforms baselines in this evaluation under different shots. It indicates that our RECORDS that considers the label disambiguation dynamic can help extract better representation.

**Other dynamic strategies.** To verify the effectiveness of RECORD, we setup two other straightforward dynamic strategies: (1) Temp Oracle-LA: temperature scaling $\mathbb{P}_{train}(y)$ from 0 to 1 during training; (2) Epoch RECORDS: use the prediction of the latest epoch to estimate the class distribution and rebalance. In Table 5, we can find that although simper solutions might be effective, RECORDS outperforms them significantly, confirming the effectiveness of our design.

**More baselines for imbalanced learning.** We additionally compare our RECORDS with two contrastive-based SOTA long-tailed learning baselines, BCL (Zhu et al., 2022) and PaCO (Cui et al., 2021), and two regularization methods for mitigating imbalance in PLL, LDD (Liu et al., 2021) and SFN (Liu et al., 2021). Similar to Oracle-LA, we use the oracle class distributions for BCL and PaCO, denoted as Oracle-BCL and Oracle-PaCO. In Table 4, RECORDS still significantly outperforms these methods, showing the effectiveness of dynamic rebalancing.

**Effect of the momentum coefficient $m$.** In Figure 6(b), we explore the effect of the momentum update factor $m$ on the performance of RECORDS. As can be seen, the best result is achieved at $m = 0.9$ and the performance decreases when $m$ takes smaller values or a larger value. Specially, when $m = 0$, it still maintains a competitive result, showing the robustness of RECORDS. At the other extreme value, *i.e.*, $m = 1.0$, CORR + RECORDS actually degenerates to CORR.

## 6 CONCLUSION

In this paper, we focus on a more practical LT-PLL scenario and identify several critical challenges in this task based on previous independent research paradigms LT and PLL. To avoid the drawback of their combination, we propose a novel method for LT-PLL, RECORDS, without requiring the prior of the oracle class distribution. Both empirical and theoretical analysis show that our proposed parametric class distribution is asymmetrically approach the static oracle class prior during training and is more friendly to label disambiguation. Our method is orthogonal to existing PLL methods and can be easily plugged into current PLL methods in an end-to-end manner. Extensive experiments demonstrate the effectiveness of our proposed RECORDS. In the future, the extension of RECORDS to other weakly supervised learning scenarios can be explored in a more general scope.

## ETHICS STATEMENT

This paper does not raise any ethics concerns. This study does not involve any human subjects, practices to data set releases, potentially harmful insights, methodologies and applications, potential conflicts of interest and sponsorship, discrimination/bias/fairness concerns, privacy and security issues, legal compliance, and research integrity issues.

## REPRODUCIBILITY STATEMENT

To ensure the reproducibility of experimental results, our code is available at `https://github.com/MediaBrain-SJTU/RECORDS-LTPLL`. We provide experimental setups and implementation details in Section 5 and Appendix E. The proof of Proposition 1 is given in Appendix D.

## ACKNOWLEDGMENTS

This work is supported by STCSM (No. 22511106101, No. 18DZ2270700, No. 21DZ1100100), 111 plan (No. BP0719010), and State Key Laboratory of UHD Video and Audio Production and Presentation.

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

# Appendix

C​ONTENTS

## A   L​ONG-T​AILED P​ARTIAL L​ABEL L​EARNING: B​ACKGROUND

The remarkable success of deep learning is built on a large amount of labeled data. Data annotation in real-world scenarios often suffers from annotation ambiguity. To address annotation ambiguity, partial label learning allows multiple candidate labels to be annotated for each training instance, which can be widely used in web mining (Luo & Orabona, 2010), automatic image annotations (Zeng et al., 2013; Chen et al., 2018), ecoinformatics (Liu & Dietterich, 2012), and crowdsourcing (Gong et al., 2018).

For example, a movie clip may contain several characters talking to each other, with some of them appearing in a screenshot. Although we can obtain scripts and dialogues that indicate the names of the characters, we cannot directly confirm the real name of each face in the screenshot (see Figure 7(a)). A similar scenario arises for recognizing faces from news images, where we can obtain the names of the people from the news descriptions but cannot establish a one-to-one correspondence with the face images (see Figure 7(b)). Partial label learning problem also appears in crowdsourcing, where each instance may be given multiple labels by different annotators. However, some labels may be incorrect or biased due to differences in expertise or cultural background of different annotators, so it is necessary to find the most appropriate label for each instance from candidate labels (see Figure 7(c)).

Meanwhile, in the real world, the data naturally exhibit a long-tailed distribution. Corresponding to the three examples in Figure 7, the main characters in movies tend to take up most of the time, while the supporting roles appear much less frequently; in sports news, superstars get most of the exposure, while many role players only get few appearances; and the number of different species in nature also shows a clear long-tailed distribution. However, such common long-tailed distribution of real data has been ignored by most existing PLL methods. Further, the invisibility of ground truth in PLL makes it difficult to even manually balance the dataset. Therefore, we focus on the more difficult but common LT-PLL scenario where label ambiguity and category imbalance co-occur.

When facing LT-PLL, directly combining the paradigms of partial label learning and long-tailed learning poses some dilemmas. One immediate problem is that the skewed long-tailed distribution exacerbates the bias toward head classes in label disambiguation and tends to lead to trivial solutions with overconfidence in head classes. More importantly, most state-of-the-art long-tailed learning methods cannot be directly used in LT-PLL because they require available class distributions, which are agnostic in PLL due to label ambiguity. Moreover, we find that existing techniques perform poorly in LT-PLL and even fail in some cases even after applying an oracle class distribution prior in the training.

## B   P​RELIMINARIES

### B.1   S​ELF-T​RAINING PLL METHODS

The self-training PLL methods (Feng et al., 2020; Wen et al., 2021; Lv et al., 2020; Fei et al., 2022) remove annotation ambiguities from model outputs and gradually identify true labels during training, achieving state-of-the-art results. These methods maintain class-wise confidence weights $w$ for each sample during training and formulate the loss function as a weighted summation of the classification

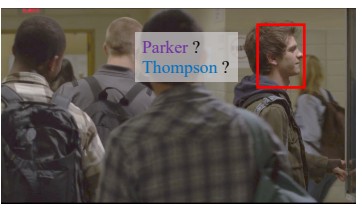
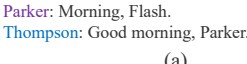
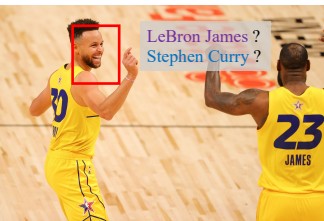
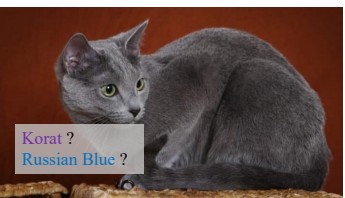

Parker: Morning, Flash.              The History of LeBron James and Stephen        Annotator 1: Korat
Thompson: Good morning, Parker.      Curry's Rivalrous Friendship                 Annotator 2: Russian Blue

          (a)                               (b)                         (c)

Figure 7: Some practical applications of PLL. (a) A screenshot of the movie "The Amazing Spider-Man" and the corresponding dialogue script. (b) An image from NBA news with the caption "The History of LeBron James and Stephen Curry's Rivalrous Friendship". (c) an image of "Russian Blue", which is very similar to "Korat".

loss by $\boldsymbol{w}$. They update the confidence weights $\boldsymbol{w}$ from the output, then use $\boldsymbol{w}$ as soft/hard pseudo-labels to learn classifiers from the data. Model training and weight updates are performed iteratively in each round. In the late training period, $w$ converges to stable one-hot labels, and PLL is converted to an ordinary classification problem that learns from the disambiguated labels. The differences among these self-training methods are the form of loss function and how the weights are updated. In Tabel 6, we summarize four state-of-the-art self-training methods, PRODEN (Lv et al., 2020), LW (Wen et al., 2021), CAVL (Fei et al., 2022), and CORR (Wu et al., 2022). Specially, PRODEN assigns higher confidence weights to labels with larger output logit; LW additionally considers the loss of non-partial labels and assigns the corresponding confidence weights to non-partial labels as well; CAVL designs a identification strategy based on the idea of CAM and assigns hard confidence weights to labels; and CORR applies consistency regularization in the disambiguation strategy.

Table 6: The loss functions and confidence weights updating strategies for self-training PLL methods.

| Methods | Per-Sample Loss | Updating $\boldsymbol{w}$ |
|---------|-----------------|---------------------------|
| PRODEN | $\sum_{j \in \mathcal{Y}} \boldsymbol{w}_{i,j} \ell(z^j(\boldsymbol{x}_i))$ | $\boldsymbol{w}_{i,j} = \frac{\exp(z^j(\boldsymbol{x}_i))}{\sum_{k \in S_i} \exp(z^k(\boldsymbol{x}_i))}$, if $j \in S_i$ 

 $\boldsymbol{w}_{i,j} = 0$, if $j \notin S_i$ |
| LW | $\sum_{j \in S_i} \boldsymbol{w}_{i,j} \ell(z^j(\boldsymbol{x}_i)) +$ 

 $\beta \sum_{j \notin S_i} \boldsymbol{w}_{i,j} \ell(-z^j(\boldsymbol{x}_i))$ | $\boldsymbol{w}_{i,j} = \frac{\exp(z^j(\boldsymbol{x}_i))}{\sum_{k \in S_i} \exp(z^k(\boldsymbol{x}_i))}$, if $j \in S_i$ 

 $\boldsymbol{w}_{i,j} = \frac{\exp(z^j(\boldsymbol{x}_i))}{\sum_{k \notin S_i} \exp(z^k(\boldsymbol{x}_i))}$, if $j \notin S_i$ |
| CAVL | $\sum_{j \in \mathcal{Y}} \boldsymbol{w}_{i,j} \ell(z^j(\boldsymbol{x}_i))$ | $\boldsymbol{w}_{i,j} = 1$, if $j = \mathrm{argmax}_{j \in S_i} \left| z^j(\boldsymbol{x}_i) - 1 \right| z^j(\boldsymbol{x}_i)$ 

 $\boldsymbol{w}_{i,j} = 0$, else |
| CORR | $\lambda D_{\mathrm{KL}}(z(\boldsymbol{x}_i) \| \boldsymbol{w}_i) +$ 

 $\sum_{j \notin S_i} \ell(-z^j(\boldsymbol{x}_i))$ | $\boldsymbol{w}_{i,j} = \frac{(\prod_{\boldsymbol{x}' \in A(\boldsymbol{x}_i)} \exp(z^j(\boldsymbol{x}')))^{\frac{1}{|A(\boldsymbol{x}_i)|}}}{\sum_{k \in \mathcal{Y}}(\prod_{\boldsymbol{x}' \in A(\boldsymbol{x}_i)} \exp(z^k(\boldsymbol{x}')))^{\frac{1}{|A(\boldsymbol{x}_i)|}}}$, if $j \in S_i$ 

 $\boldsymbol{w}_{i,j} = 0$, if $j \notin S_i$ |

## B.2 LOGIT ADJUSTMENT FOR LONG-TAILED LEARNING

Here we give a review on the successful logit adjustment (LA) (Menon et al., 2021; Hong et al., 2021) in long-tailed learning. LA uses the Bayes Rule to remove the adverse effect of class imbalance by performing a offset term on the output logit. In the long-tailed setting with a highly skewed

training category distribution $\mathbb{P}_{train}(y)$, a trivial classifier that classifies all instances as majority labels can achieve high training accuracy. To cope with it, a class-balanced test set $\mathcal{D}_{uni} = \{(\boldsymbol{x}_i, y_i)\}$ is used (Brodersen et al., 2010). We donate $\mathbb{P}_{uni}(y|\boldsymbol{x}) \propto \mathbb{P}(\boldsymbol{x}|y)$ as the underlying class probability under uniform distribution $\mathbb{P}_{uni}(y) = \frac{1}{C}$. The of long-tailed learning is to learn a model on $\mathcal{D}_{train}$ that minimizes the following balanced error rate (BER) on $\mathcal{D}_{uni}$:

$$\min_{\Theta} \mathrm{BER}(\Theta) \stackrel{\mathbb{P}_{uni}(y)=\frac{1}{C}}{=} \mathbb{P}_{(\boldsymbol{x},y)\in\mathcal{D}_{uni}}(y \neq \operatorname*{argmax}_{y' \in \mathcal{Y}} z^{y'}(\boldsymbol{x})).$$

For an optimal model trained by minimizing the traditional softmax cross-entropy loss, the output logit $z$ satisfies $\mathrm{softmax}(z^y(\boldsymbol{x})) = \mathbb{P}_{train}(y|\boldsymbol{x}) \propto \mathbb{P}(\boldsymbol{x}|y) \cdot \mathbb{P}_{train}(y)$ (Yu et al., 2018). Recall that the goal is to minimize BER. For a BER-optimal model, also known as Bayes-optimal model, its output logit $z_{uni}$ satisfies $\mathrm{softmax}(z^y_{uni}(\boldsymbol{x})) = \mathbb{P}_{uni}(y|\boldsymbol{x})$ (Menon et al., 2013). LA attempts to convert the output $z$ obtained from traditional cross-entropy training into a Bayes-optimal output as follows:

$$\mathrm{softmax}(z^y(\boldsymbol{x})) = \mathbb{P}_{train}(y|\boldsymbol{x}) \propto \mathbb{P}(\boldsymbol{x}|y) \cdot \mathbb{P}_{train}(y)$$
$$\mathbb{P}_{uni}(y|\boldsymbol{x}) \propto \mathbb{P}(\boldsymbol{x}|y) \propto \mathrm{softmax}(z^y(\boldsymbol{x}) - \log \mathbb{P}_{train}(y)).$$

That is to say, if we have the logit $z^y(\boldsymbol{x})$ by training on standard cross-entropy loss, a Bayes-optimal logit $z^y_{uni}(\boldsymbol{x}) = z^y(\boldsymbol{x}) - \log \mathbb{P}_{train}(y)$ can be obtained by subtracting an offset term $\log \mathbb{P}_{train}(y)$. Using $z^y_{uni}(\boldsymbol{x})$ as the test logit preserves the balanced part $\mathbb{P}(\boldsymbol{x}|y)$ of the output and removes the imbalanced part $\mathbb{P}_{train}(y)$ in a statistical sense. A number of recent studies (Zhu et al., 2022; Cui et al., 2021) have demonstrated the powerful effects of logit adjustment in long-tailed learning.

## C  PSEUDO-CODE OF RECORDS

We summarize the complete procedure of our RECORDS in Algorithm 1.

---
**Algorithm 1:** Our proposed RECORDS.

---
**Input:** Training dataset $\mathcal{D}_{train}$, deep model $f$, classifier $g$, a self-training PLL algorithm $\mathcal{A}$,
      number of epochs $T$.
**Output:** Parameter $\theta$ for f, parameter $\boldsymbol{W}$ for g.

1 Initialize uniform weights $\boldsymbol{w}$;
2 Initialize $F$ with 0;
3 **for** $t = 1, 2, \dots, T$ **do**
4      Shuffle training set $\mathcal{D}_{train}$ into $B$ mini-batches;
5      **for** $k = 1, 2, \dots, B$ **do**
6          Compute output $z$ for mini-batch $\mathcal{D}_k$;
7          Update $F$ according to Equation 5;
8          Compute loss $L_{\mathcal{A}}$ according to algorithm $\mathcal{A}$;
9          Compute debiased output $z_{uni}$ according to Equation 6;
10         Update $\boldsymbol{w}$ using $z_{uni}$ according to algorithm $\mathcal{A}$;
11         Update $\theta$ and $\boldsymbol{W}$ by minimizing $L_{\mathcal{A}}$;
12      **end**
13 **end**

---

## D  PROOF OF PROPOSITION 1

**Lemma D.1.** *Let $L_2(h)$ for $h \in H$ be $L_2$ distance between $\mathbb{P}_{train}(y)$ and $\mathbb{P}_{train}(y|\Theta)$, where $h$ is parameterized by $\Theta$. We have*

$$L_2(h) \leq 2\mathbb{E}_{(\boldsymbol{x},y)\sim(\mathcal{X},\mathcal{Y})}\mathbb{1}(h(\boldsymbol{x}) \neq y)).$$

*Proof.*

$$L_2(h) = \sqrt{\sum_{y_j=1}^{C} \left| \mathbb{E}_{(\boldsymbol{x},y)\sim(\mathcal{X},\mathcal{Y})}(\mathbb{1}(y = y_j) - \mathbb{1}(h(\boldsymbol{x}) = y_j)) \right|^2} \tag{7}$$

$$\leq \sum_{y_j=1}^{C} \mathbb{E}_{(\boldsymbol{x},y)\sim(\mathcal{X},\mathcal{Y})} \left| \mathbb{1}(y = y_j) - \mathbb{1}(h(\boldsymbol{x}) = y_j) \right|$$

For any instance $(\boldsymbol{x}, y)$, if $h(\boldsymbol{x}) = y$, *i.e.,* $h$ correctly classifies $\boldsymbol{x}$, then for $\forall y_j \in \{1, 2, \ldots, C\}$, we have $|\mathbb{1}(y = y_j) - \mathbb{1}(h(\boldsymbol{x}) = y_j)| = 0$. Otherwise if $h(\boldsymbol{x}) \neq y$, we have

$$|\mathbb{1}(y = y_j) - \mathbb{1}(h(\boldsymbol{x}) = y_j)| = \begin{cases} 1, & y_j = y \text{ or } y_j = h(\boldsymbol{x}) \\ 0, & else \end{cases} \tag{8}$$

Thus, we can bound $L_2(h)$ by

$$L_2(h) \leq 2\mathbb{E}_{(\boldsymbol{x},y)\sim(\mathcal{X},\mathcal{Y})}\mathbb{1}(h(\boldsymbol{x}) \neq y)) \tag{9}$$

$\square$

**Lemma D.2.** *Define $R$ as set of all $\mathcal{D}_{train}$ for which there exists an $\epsilon$-$L_2$ Distance $h$ with zero empirical risk: $R = \{\mathcal{D}_{train} \in (\mathcal{X}, \mathcal{Y}, \mathcal{S})^N : \exists h, L_2(h) \geq \epsilon, R^{\mathcal{D}_{train}}(h) = 0\}$. Introduce another training set $\mathcal{D}'_{train} \in (\mathcal{X}, \mathcal{Y}, \mathcal{S})^N$. Define $M$ to be set of all pairs $(\mathcal{D}_{train}, \mathcal{D}'_{train})$ for which there exists a $h$ with zero empirical risk on $\mathcal{D}_{train}$ that makes $\epsilon$-$L_2$ Distance and make at least $\frac{\epsilon N}{4}$ classification errors on $\mathcal{D}'_{train}$, i.e., $M = \{(\mathcal{D}_{train}, \mathcal{D}'_{train}) \in (\mathcal{X}, \mathcal{Y}, \mathcal{S})^{2N} : \exists h, L_2(h) \geq \epsilon, R^{\mathcal{D}_{train}}(h) = 0, \sum_{i=1}^{N} \mathbb{1}(h(\boldsymbol{x}') \neq y') \geq \frac{\epsilon N}{4}\}$. If $N > \frac{16 \ln 2}{\epsilon}$, then we have*

$$\mathbb{P}(\mathcal{D}_{train} \in R) < 2\mathbb{P}((\mathcal{D}_{train}, \mathcal{D}'_{train}) \in M).$$

*Proof.* This lemma is used in many learnability proofs. Apply the chain rule of probability:

$$\mathbb{P}((\mathcal{D}_{train}, \mathcal{D}'_{train}) \in M)$$
$$=\mathbb{P}((\mathcal{D}_{train}, \mathcal{D}'_{train}) \in M | \mathcal{D}_{train} \in R) \cdot \mathbb{P}(\mathcal{D}_{train} \in R)$$
$$=\mathbb{P}(\exists h, L_2(h) \geq \epsilon, R^{\mathcal{D}_{train}}(h) = 0, \sum_{i=1}^{N} \mathbb{1}(h(\boldsymbol{x}') \neq y') \geq \frac{\epsilon N}{4} | \mathcal{D}_{train} \in R) \cdot \mathbb{P}(\mathcal{D}_{train} \in R)$$
$$\geq \mathbb{P}(\sum_{i=1}^{N} \mathbb{1}(h(\boldsymbol{x}') \neq y') \geq \frac{\epsilon N}{4} | L_2(h) \geq \epsilon, R^{\mathcal{D}_{train}}(h) = 0) \cdot \mathbb{P}(\mathcal{D}_{train} \in R)$$

$$\tag{10}$$

Given $L_2(h) \geq \epsilon$, we can get $\mathbb{E}_{(\boldsymbol{x},y)\sim(\mathcal{X},\mathcal{Y})}\mathbb{1}(h(\boldsymbol{x}) \neq y) \geq \frac{\epsilon}{2}$ (Lemma D.1). Using the Chernoff bound, when $N > \frac{16 \ln 2}{\epsilon}$, we can get

$$\mathbb{P}(\sum_{i=1}^{N} \mathbb{1}(h(\boldsymbol{x}') \neq y') \geq \frac{\epsilon N}{4})$$
$$=1 - \mathbb{P}(\sum_{i=1}^{N} \mathbb{1}(h(\boldsymbol{x}') \neq y') < \frac{\epsilon N}{4})$$
$$\geq 1 - \mathbb{P}(\sum_{i=1}^{N} \mathbb{1}(h(\boldsymbol{x}') \neq y') < (1 - \frac{1}{2})\mathbb{E}_{(\boldsymbol{x}',y')\sim(\mathcal{X},\mathcal{Y})} \sum_{i=1}^{N} \mathbb{1}(h(\boldsymbol{x}') \neq y')) \tag{11}$$
$$\geq 1 - e^{-\frac{N\mathbb{E}_{(\boldsymbol{x},y)\sim(\mathcal{X},\mathcal{Y})}\mathbb{1}(h(\boldsymbol{x})\neq y)}{8}}$$
$$\geq 1 - e^{-\frac{N\epsilon}{16}} > \frac{1}{2}$$

Thus we can bound $\mathbb{P}(\mathcal{D}_{train} \in R)$ by

$$\mathbb{P}(\mathcal{D}_{train} \in R) < 2\mathbb{P}((\mathcal{D}_{train}, \mathcal{D}'_{train}) \in M) \quad (12)$$

$\square$

**Lemma D.3.** *(Liu & Dietterich, 2014) If the hypothesis space $H$ has Natarajandimension $d_H$ and $\eta < 1$, then*

$$\mathbb{P}((\mathcal{D}_{train}, \mathcal{D}'_{train}) \in M) < (2N)^{d_{\mathcal{H}}} C^{2d_{\mathcal{H}}} \left(\frac{1+\eta}{2}\right)^{\frac{\epsilon N}{4}}$$

.

*Proof.* Expand $\mathcal{D}_{train}$ and $\mathcal{D}'_{train}$ as $\mathcal{D}_{train} = (\boldsymbol{x}^N, y^N, S^N) \in (\mathcal{X}, \mathcal{Y}, \mathcal{S})^N, \mathcal{D}'_{train} = (\boldsymbol{x}'^N, y'^N, S'^N) \in (\mathcal{X}, \mathcal{Y}, \mathcal{S})^N$. We can get

$$\mathbb{P}((\mathcal{D}_{train}, \mathcal{D}'_{train}) \in M) = \mathbb{E} \, \mathbb{P}((\mathcal{D}_{train}, \mathcal{D}'_{train}) \in M | \boldsymbol{x}^N, y^N, \boldsymbol{x}'^N, y'^N) \quad (13)$$

Where the expectation $\mathbb{E}$ is taken with respect to $(\boldsymbol{x}^N, y^N, \boldsymbol{x}'^N, y'^N)$ and the probability $\mathbb{P}$ comes from $(S^N, S'^N)$ given $(\boldsymbol{x}^N, y^N, \boldsymbol{x}'^N, y'^N)$.

Let $H|(\boldsymbol{x}^N, \boldsymbol{x}'^N)$ be the set of hypothesis making different classifications for $\mathcal{D}_{train}$ and $\mathcal{D}'_{train}$, we can get

$$\mathbb{P}((\mathcal{D}_{train}, \mathcal{D}'_{train}) \in M | \boldsymbol{x}^N, y^N, \boldsymbol{x}'^N, y'^N)$$
$$\leq \sum_{h \in H|(\boldsymbol{x}^N, \boldsymbol{x}'^N)} \mathbb{P}(R^{\mathcal{D}_{train}}(h) = 0, \sum_{i=1}^{N} \mathbb{1}(h(\boldsymbol{x}') \neq y') \geq \frac{\epsilon N}{4} | \boldsymbol{x}^N, y^N, \boldsymbol{x}'^N, y'^N) \quad (14)$$

$$\mathbb{P}((\mathcal{D}_{train}, \mathcal{D}'_{train}) \in M | \boldsymbol{x}^N, y^N, \boldsymbol{x}'^N, y'^N)$$
$$\leq \sum_{h \in H|(\boldsymbol{x}^N, \boldsymbol{x}'^N)} \mathbb{P}(R^{\mathcal{D}_{train}}(h) = 0, \sum_{i=1}^{N} \mathbb{1}(h(\boldsymbol{x}') \neq y') \geq \frac{\epsilon N}{4} | \boldsymbol{x}^N, y^N, \boldsymbol{x}'^N, y'^N) \quad (15)$$

Randomly match the instances in $\mathcal{D}_{train}$ and $\mathcal{D}'_{train}$ into $n$ pairs. Following Liu & Dietterich (2014), we define a group $G$ of swaps. A swap $\sigma \in G$ swaps the instances in pairs indexed by $J_\sigma \subseteq \{1, 2, \ldots, n\}$. $|G| = 2^N, \sigma(\mathcal{D}_{train}, \mathcal{D}'_{train}) = (\mathcal{D}^\sigma_{train}, \mathcal{D}'^\sigma_{train})$. Let $a_1, a_2$ be the number for which $h$ classifies $\mathcal{D}_{train}$ incorrectly, and $\mathcal{D}'_{train}$ incorrectly. Let $b_1, b_2$ be the number for which $h$ classifies both incorrectly, and only one incorrectly.

$$\mathbb{P}(R^{\mathcal{D}_{train}}(h) = 0, \sum_{i=1}^{N} \mathbb{1}(h(\boldsymbol{x}') \neq y') \geq \frac{\epsilon N}{4} | \boldsymbol{x}^N, y^N, \boldsymbol{x}'^N, y'^N)$$
$$= \mathbb{1}(\sum_{i=1}^{N} \mathbb{1}(h(\boldsymbol{x}') \neq y') \geq \frac{\epsilon N}{4}) \cdot \prod_{i=1}^{N} \mathbb{P}(h(\boldsymbol{x}_i) = \tilde{y}(\boldsymbol{x}_i, S_i) | \boldsymbol{x}^N, y^N, \boldsymbol{x}'^N, y'^N) \quad (16)$$
$$= \mathbb{1}(a_2 \geq \frac{\epsilon N}{4}) \cdot \prod_{i=1}^{N} \mathbb{P}(h(\boldsymbol{x}_i) = \tilde{y}(\boldsymbol{x}_i, S_i) | \boldsymbol{x}^N, y^N, \boldsymbol{x}'^N, y'^N)$$

We then give the bound for $\prod_{i=1}^{N} \mathbb{P}(h(\boldsymbol{x}_i) = \tilde{y}(\boldsymbol{x}_i, S_i) | \boldsymbol{x}^N, y^N, \boldsymbol{x}'^N, y'^N)$.

$$\prod_{i=1}^{N} \mathbb{P}(h(\boldsymbol{x}_i) = \tilde{y}(\boldsymbol{x}_i, S_i) | \boldsymbol{x}^N, y^N, \boldsymbol{x}'^N, y'^N) \leq \prod_{i=1}^{N} \mathbb{P}(h(\boldsymbol{x}_i) \in S_i | \boldsymbol{x}^N, y^N, \boldsymbol{x}'^N, y'^N) \leq \eta^{a_1} \quad (17)$$

Combining Equation 13-17, we can get

$$
\begin{aligned}
&\mathbb{P}((\mathcal{D}_{train}, \mathcal{D}'_{train}) \in M) \\
&= \mathbb{E}\, \mathbb{P}((\mathcal{D}_{train}, \mathcal{D}'_{train}) \in M | \boldsymbol{x}^N, y^N, \boldsymbol{x}'^N, y'^N) \\
&= \mathbb{E}\, \frac{1}{2^N} \sum_{\sigma \in G} \mathbb{P}(\sigma(\mathcal{D}_{train}, \mathcal{D}'_{train}) \in M | \boldsymbol{x}^N, y^N, \boldsymbol{x}'^N, y'^N) \\
&\leq \mathbb{E}\, \frac{1}{2^N} \sum_{\sigma \in G} \sum_{h \in H | \sigma(x, \boldsymbol{x}')} \mathbb{P}(R^{\mathcal{D}_{train}}(h) = 0, \sum_{i=1}^{N} \mathbb{1}(h(\boldsymbol{x}') \neq y') \geq \frac{\epsilon N}{4} | \boldsymbol{x}^N, y^N, \boldsymbol{x}'^N, y'^N) \\
&\leq \mathbb{E}\, (2N)^{d_{\mathcal{H}}} C^{2 d_{\mathcal{H}}} \frac{1}{2^N} \sum_{\sigma \in G} \mathbb{1}(a_2^\sigma \geq \frac{\epsilon N}{4}) \eta^{a_1^\sigma} \\
&\leq \mathbb{E}\, (2N)^{d_{\mathcal{H}}} C^{2 d_{\mathcal{H}}} \frac{1}{2^N} \sum_{a_1^\sigma = b_1}^{b_1 + b_2} \mathbb{1}(b_1 + b_2 \geq \frac{\epsilon N}{4}) C_{b_2}^{a_1^\sigma - b_1} 2^{n - b_2} \eta^{a_1^\sigma} \\
&= \mathbb{E}\, (2N)^{d_{\mathcal{H}}} C^{2 d_{\mathcal{H}}} \mathbb{1}(b_1 + b_2 \geq \frac{\epsilon N}{4}) \eta^{b_1} (\frac{1 + \eta}{2})^{b_2}
\end{aligned}
\tag{18}
$$

where the third line uses the inequality $|H|(\boldsymbol{x}^N, \boldsymbol{x}'^N)| \leq (2N)^{d_{\mathcal{H}}} C^{2 d_{\mathcal{H}}}$ (Natarajan, 1989). When $b_1 = 0, b_2 = \frac{\epsilon N}{4}$, the right side reaches the maximum of $(2N)^{d_{\mathcal{H}}} C^{2 d_{\mathcal{H}}} (\frac{1 + \eta}{2})^{\frac{\epsilon N}{4}}$. $\qquad \square$

For ease of reading, here we restate Proposition 1 in the main text.

**Proposition D.1.** *Let* $\eta = \sup_{(\boldsymbol{x}, y) \in \mathcal{X} \times \mathcal{Y}, y_j \in \mathcal{Y}, y_j \neq y} \mathbb{P}_{S | (\boldsymbol{x}, y)}(y_j \in S)$ *denote the ambiguity degree,* $d_H$ *be the Natarajan dimension of the hypothesis space* $H$, $\tilde{h} = h_{\tilde{\Theta}}$ *be the optimal classifier on the basis of the label disambiguation, where* $\tilde{\Theta} = \arg\min_\Theta R^{\mathcal{D}_{train}}(\Theta)$. *If the small ambiguity degree condition (Cour et al., 2011a; Liu & Dietterich, 2014)) satisfies, namely,* $\eta \in [0, 1)$, *then for* $\forall \delta > 0$, *the* $L_2$ *distance between* $\mathbb{P}_{train}(y)$ *and* $\mathbb{P}_{train}(y | \tilde{\Theta})$ *for* $\tilde{h}$ *is bounded as*

$$
L_2(\tilde{h}) < \frac{4}{(\ln 2 - \ln(1 + \eta)) N} (d_H (\ln 2N + 2 \ln C) - \ln \delta + \ln 2)
$$

*with probability at least* $1 - \delta$, *where* $N$ *is the sample number and* $C$ *is the category number.*

*Proof.* Recall the definition of $R$:

$$
R = \{\mathcal{D}_{train} \in (\mathcal{X}, \mathcal{Y}, \mathcal{S})^N : \exists h, L_2(h) \geq \epsilon, R^{\mathcal{D}_{train}}(h) = 0\}
\tag{19}
$$

Here we need to prove the sufficient condition for $\mathbb{P}(\mathcal{D}_{train} \in R) \leq \delta$. Lemma D.2 bounds $\mathbb{P}(\mathcal{D}_{train} \in R)$ by $\mathbb{P}(\mathcal{D}_{train} \in R) < 2\mathbb{P}((\mathcal{D}_{train}, \mathcal{D}'_{train}) \in M)$. Lemma D.3 bounds $\mathbb{P}((\mathcal{D}_{train}, \mathcal{D}'_{train}) \in M)$ by $(2N)^{d_{\mathcal{H}}} C^{2 d_{\mathcal{H}}} (\frac{1 + \eta}{2})^{\frac{\epsilon N}{4}}$. Taking it into Lemma D.2, we get

$$
\mathbb{P}(\mathcal{D}_{train} \in R) < 2\mathbb{P}((\mathcal{D}_{train}, \mathcal{D}'_{train}) \in M) \leq 2(2N)^{d_{\mathcal{H}}} C^{2 d_{\mathcal{H}}} (\frac{1 + \eta}{2})^{\frac{\epsilon N}{4}}
\tag{20}
$$

Then we can prove the sufficient condition for $\mathbb{P}(\mathcal{D}_{train} \in R) \leq \delta$:

$$
\begin{aligned}
\mathbb{P}(\mathcal{D}_{train} \in R) \leq \delta \impliedby\ & \ln 2 + d_{\mathcal{H}} (\ln 2 + \ln N) + 2 d_{\mathcal{H}} \ln C - \frac{\epsilon N}{4} \ln(\frac{1 + \eta}{2}) \leq \ln \delta \\
\iff\ & \epsilon \geq \frac{4}{(\ln 2 - \ln(1 + \eta)) N} (d_H (\ln 2N + 2 \ln C) - \ln \delta + \ln 2)
\end{aligned}
\tag{21}
$$

That is, when $\epsilon = \frac{4}{(\ln 2 - \ln(1 + \eta)) N} (d_H (\ln 2N + 2 \ln C) - \ln \delta + \ln 2)$, we have $L_2\left(\tilde{h}\right) < \epsilon$ with probability at least $1 - \delta$.

$\qquad \square$

Table 7: Characteristics of PASCAL VOC

| #Classes | #Train | #Test | Imbalance ratio | Avg #Candidate Labels |
|---|---|---|---|---|
| 20 | 11706 | 4000 | 118.8 | 2.46 |

Table 8: Sample size for each class of PASCAL VOC

| Class | person | chair | car | bottle | sofa | bicycle | horse | pottedplant | diningtable | motorbike |
|---|---|---|---|---|---|---|---|---|---|---|
| Number | 5702 | 1285 | 884 | 452 | 364 | 354 | 333 | 319 | 304 | 300 |
| Class | tvmonitor | dog | bus | cow | boat | bird | cat | train | sheep | aeroplane |
| Number | 277 | 251 | 193 | 134 | 125 | 117 | 111 | 87 | 66 | 48 |

## E  ADDITIONAL EXPERIMENTAL SETUP AND RESULTS

### E.1  DATASETS

**CIFAR-10-LT and CIFAR-100-LT.** The original versions of CIFAR-10 and CIFAR-100 contain 50,000 training images and 10,000 validation images in $32 \times 32$ size with 10 and 100 categories, respectively. The 100 categories in CIFAR-100 form 20 superclasses, each with 5 classes. Following Liu et al. (2019), we conduct the long-tailed version of CIFAR-10 and CIFAR-100 by an exponential decay across sample sizes of different classes, namely CIFAR-10-LT and CIFAR-100-LT. The imbalance rate $\rho$ denotes the ratio of the sample sizes of the most frequent and least frequent classes, *i.e.,* $\rho = \frac{N_C}{N_1}$, where the sample number $N_c$ of each class $c \in \mathcal{Y}$ is in the descending order.

*Candidate label sets generation.* Following Lv et al. (2020); Wen et al. (2021), we adopt the *uniform* setting in CIFAR-10-LT and CIFAR-100-LT, *i.e.,* $\mathbb{P}(\overline{y} \in S | \overline{y} \neq y) = q$ to generate candidate label set of each sample. That is, all the $C - 1$ negative labels have the same probability of $q$ to be selected into the candidate set along with the ground truth. Besides, for CIFAR-100-LT, we also use a more challenging *non-uniform* setting where other labels in the same superclass of the ground truth have a higher probability to be selected into the candidate set, *i.e.,* $\mathbb{P}(\overline{y} \in S | \overline{y} \neq y, D(\overline{y}) \neq D(y)) = q, \mathbb{P}(\overline{y} \in S | \overline{y} \neq y, D(\overline{y}) = D(y)) = 8q$, where $D(y)$ denotes the superclass to which $y$ belongs. The non-uniform setting is more practical and challenging for the LT-PLL algorithms because the semantics of the categories in the same superclass are closer. We refer to CIFAR-100-LT with candidates generated by the non-uniform setting as CIFAR-100-LT-NU.

*Dataset partitioning.* To demonstrate the performance of the algorithm on categories with different frequencies, we partition the dataset according to the sample size. Following Kang et al. (2020), We split the dataset into three partitions: Many-shot (classes with more than 100 images), Medium-shot (classes with 20-100 images), and Few-shot (classes with less than 20 images).

**PASCAL VOC.** We conduct the real-world LT-PLL dataset PASCAL VOC from PASCAL VOC 2007 (Everingham et al.). Specifically, we crop objects in images as instances and all objects appearing in the same original image are regarded as the labels of a candidate set. Note that, here we are the first to conduct experiments based on deep models on real-world datasets, while previous PLL real-world datasets are tabular and only suitable for linear models or shallow MLPs (Fei et al., 2022). Empirically we can observe the significant class imbalance as shown in Table 8. We manually balanced the test set for fairness of the evaluation. In Table 7, we summarize the basic characteristics of PASCAL VOC.

### E.2  ADDITIONAL IMPLEMENTATION DETAILS

**Toy study.** (Figure 4) We simulated a four-class LT-PLL task with each class distributed in a different region and their samples uniformly distributed in the corresponding region. The four uniform distributions are $\mathbb{U}([-1, -1], [0, 0])$, $\mathbb{U}([0, -1], [1, 0])$, $\mathbb{U}([-1, 0], [0, 1])$, and $\mathbb{U}([0, 0], [1, 1])$, respectively. The candidate label set for each sample consists of the ground truth and negative labels with 0.6 probability of being flipped from the other classes *e.g.,* $\mathbb{P}(\overline{y} \in S | \overline{y} \neq y) = 0.6$. The sample size of each class in the training set is 30 (red), 100 (yellow), 500 (blue) and 1000 (green), respectively. The test set is balanced, with 100 samples per class. We use a two-layer MLP with 10 hidden

Table 9: Top-1 accuracy on CIFAR-100-LT-NU with imbalance ratio $\rho \in \{50, 100\}$ and ambiguity $q \in \{0.03, 0.05, 0.07\}$. Bold indicates superior results.

| Imbalance ratio $\rho$ | 50 | | | 100 | | |
|---|---|---|---|---|---|---|
| ambiguity $q$ | 0.03 | 0.05 | 0.07 | 0.03 | 0.05 | 0.07 |
| CORR | 40.73 | 35.76 | 32.21 | 36.39 | 32.17 | 28.56 |
| + Oracle-LA post-hoc | 45.57 | 39.07 | 34.75 | 39.95 | 35.38 | 31.44 |
| + Oracle-LA | 16.52 | 10.25 | 8.61 | 8.56 | 5.85 | 5.92 |
| + RECORDS | **46.98** | **44.93** | **43.37** | **40.93** | **39.36** | **37.06** |
| vs. CORR | +6.25 | +9.17 | +11.16 | +4.54 | +7.19 | +8.50 |
| PRODEN | 38.00 | 33.03 | 28.29 | 32.97 | 29.98 | 26.45 |
| + Oracle-LA post-hoc | 42.69 | 35.86 | 30.42 | 36.83 | 32.72 | 29.07 |
| + Oracle-LA | 11.44 | 7.28 | 6.50 | 6.65 | 5.23 | 4.96 |
| + RECORDS | **43.83** | **41.39** | **39.88** | **37.99** | **35.76** | **34.19** |
| vs. PRODEN | +5.83 | +8.36 | +11.59 | +5.02 | +5.78 | +7.74 |
| LW | 33.76 | 28.73 | 26.11 | 30.10 | 25.56 | 22.58 |
| + Oracle-LA post-hoc | 33.53 | 28.39 | 25.83 | 29.31 | 24.56 | 21.84 |
| + Oracle-LA | 28.22 | 12.13 | 11.11 | 14.36 | 7.73 | 6.85 |
| + RECORDS | **34.93** | **30.78** | **27.79** | **31.51** | **26.65** | **25.06** |
| vs. LW | +1.17 | +2.05 | +1.68 | +1.41 | +1.09 | +2.48 |
| CAVL | 27.30 | 20.85 | 13.18 | 27.26 | 20.53 | 7.34 |
| + Oracle-LA post-hoc | 20.56 | 14.96 | 9.88 | 20.52 | 13.56 | 4.93 |
| + Oracle-LA | 11.66 | 7.11 | 6.47 | 6.86 | 5.16 | 4.80 |
| + RECORDS | **39.71** | **31.02** | **18.42** | **34.77** | **26.36** | **20.45** |
| vs. CAVL | +12.41 | +10.17 | +5.24 | +7.51 | +5.83 | +13.11 |

Table 10: Fine-grained analysis on CIFAR-100-LT with $\rho = 100$ and $q \in \{0.03, 0.05, 0.07\}$. Many/Medium/Few corresponds to three partitions on the long-tailed data. Bold indicates superior results.

| Method | $q = 0.03$ | | | | $q = 0.05$ | | | | $q = 0.07$ | | | |
|---|---|---|---|---|---|---|---|---|---|---|---|---|
| | Many | Medium | Few | Overall | Many | Medium | Few | Overall | Many | Medium | Few | Overall |
| CORR | 68.43 | 37.40 | 4.50 | 38.39 | 67.51 | 29.60 | 0.33 | 34.09 | 68.86 | 19.80 | 0.07 | 31.05 |
| + Oracle-LA post-hoc | 70.37 | 41.89 | 7.33 | 41.49 | 70.46 | 33.40 | 1.47 | 36.79 | 69.77 | 24.86 | 0.67 | 33.32 |
| + Oracle-LA | 11.03 | 12.34 | 10.63 | 11.37 | 0.34 | 4.46 | 5.47 | 3.32 | 0.00 | 0.71 | 5.77 | 1.98 |
| + RECORDS | 66.37 | 42.54 | 13.77 | **42.25** | 68.49 | 40.20 | 8.50 | **40.59** | 69.97 | 36.71 | 4.37 | **38.65** |
| vs. CORR | -2.06 | +5.14 | +9.27 | +3.86 | +0.98 | +10.60 | +8.17 | +6.50 | +1.11 | +16.91 | +4.30 | +7.60 |
| PRODEN | 64.23 | 32.49 | 2.23 | 34.52 | 62.60 | 28.66 | 0.33 | 32.04 | 65.49 | 18.51 | 0.00 | 29.40 |
| + Oracle-LA post-hoc | 66.63 | 38.03 | 5.93 | 38.40 | 66.86 | 31.80 | 2.23 | 35.20 | 66.60 | 24.17 | 0.50 | 31.92 |
| + Oracle-LA | 2.40 | 9.31 | 8.97 | 6.79 | 0.06 | 1.74 | 7.00 | 2.73 | 0.03 | 0.20 | 6.33 | 1.98 |
| + RECORDS | 64.91 | 38.40 | 9.90 | **39.13** | 65.94 | 36.54 | 4.53 | **37.23** | 66.14 | 33.03 | 1.83 | **35.26** |
| vs. PRODEN | +0.68 | +5.91 | +7.67 | +4.61 | +3.34 | +7.88 | +4.20 | +5.19 | +0.65 | +14.52 | +1.83 | +5.86 |
| LW | 64.37 | 25.86 | 0.00 | 31.58 | 63.85 | 16.40 | 0.00 | 28.09 | 62.31 | 8.11 | 0.00 | 24.65 |
| + Oracle-LA post-hoc | 59.89 | 25.57 | 3.73 | 31.03 | 58.03 | 16.26 | 3.20 | 26.96 | 55.14 | 8.49 | 3.10 | 23.20 |
| + Oracle-LA | 50.37 | 29.60 | 7.70 | 30.30 | 4.29 | 4.63 | 6.53 | 5.08 | 0.00 | 1.54 | 7.20 | 2.70 |
| + RECORDS | 61.66 | 29.29 | 4.42 | **33.00** | 63.17 | 16.26 | 3.45 | **28.85** | 58.51 | 11.74 | 3.25 | **25.64** |
| vs. LW | -2.71 | +3.43 | +4.42 | +1.42 | -0.68 | -0.14 | +3.45 | +0.76 | -3.80 | +3.63 | +3.25 | +0.99 |
| CAVL | 59.54 | 20.80 | 0.57 | 28.29 | 58.57 | 12.97 | 0.00 | 25.39 | 16.71 | 6.65 | 0.13 | 8.20 |
| + Oracle-LA post-hoc | 51.51 | 27.26 | 2.57 | 28.34 | 57.85 | 16.51 | 0.80 | 26.27 | 7.60 | 5.54 | 3.97 | 5.80 |
| + Oracle-LA | 5.29 | 9.00 | 7.47 | 7.24 | 0.00 | 2.03 | 6.13 | 2.55 | 0.00 | 0.29 | 6.43 | 2.03 |
| + RECORDS | 62.66 | 35.43 | 8.67 | **36.93** | 54.20 | 31.43 | 5.07 | **31.49** | 44.66 | 25.49 | 1.43 | **24.98** |
| vs. CAVL | +3.12 | +14.63 | +8.10 | +8.64 | -4.37 | +18.46 | +5.07 | +6.10 | +27.95 | +18.84 | +1.30 | +16.78 |

neurons as the model for the toy study. The batch size is set to 512. The toy models are trained using SGD with momentum of 0.9. We train the models for 50 epochs with initial learning rate 2.0.

**Linear Probing.** (Figure 6(a)) We train a linear classifier on a frozen pre-trained backbone and measure the quality of the representation through the test accuracy. To eliminate the effect of the long-tailed distribution in the fine-tuning phase, the classifier is trained on a balanced dataset. Specifically, the performance of the classifier is reported on the basis of pre-trained representations for different amounts of data, including full-shot, 100-shot, and 50-shot. In the fine-tuning phase, we train the linear classifier for 500 epochs with SGD of momentum 0.7 and weight decay 0.0005. The batch size is set to 1000. The learning rate decays exponentially from $10^{-2}$ to $10^{-6}$. The loss function is set to the ordinary cross-entropy loss.

### E.3 ADDITIONAL RESULTS FOR NON-UNIFORM CANDIDATES GENERATION.

In Table 9, we show the complete experimental results on CIFAR-100-LT-NU. Under a more practical and challenging candidate generation setting, our RECORDS achieves a consistent and signifi-

Table 11: Fine-grained analysis on CIFAR-100-LT-NU with $\rho = 100$ and $q \in \{0.03, 0.05, 0.07\}$. Many/Medium/Few corresponds to three partitions on the long-tailed data. Bold indicates superior results.

| Method | $q = 0.03$ | | | | $q = 0.05$ | | | | $q = 0.07$ | | | |
|---|---|---|---|---|---|---|---|---|---|---|---|---|
| | Many | Medium | Few | Overall | Many | Medium | Few | Overall | Many | Medium | Few | Overall |
| CORR | 66.71 | 34.57 | 3.13 | 36.39 | 70.23 | 21.57 | 0.13 | 32.17 | 65.37 | 16.11 | 0.13 | 28.56 |
| + Oracle-LA post-hoc | 69.80 | 40.43 | 4.57 | 39.95 | 71.20 | 28.77 | 1.30 | 35.38 | 68.71 | 20.17 | 1.10 | 31.44 |
| + Oracle-LA | 3.26 | 9.97 | 13.10 | 8.56 | 0.86 | 5.34 | 12.27 | 5.85 | 0.00 | 0.25 | 16.87 | 5.92 |
| + RECORDS | 65.60 | 41.26 | 11.77 | **40.93** | 66.40 | 38.71 | 8.57 | **39.36** | 62.09 | 37.40 | 7.47 | **37.06** |
| vs. CORR | -1.11 | +6.69 | +8.44 | +4.54 | -3.83 | +17.14 | +8.44 | +7.19 | -3.28 | +21.29 | +7.34 | +8.50 |
| PRODEN | 63.94 | 29.74 | 0.6 | 32.97 | 65.54 | 20.09 | 0.03 | 29.98 | 62.89 | 12.69 | 0.00 | 26.45 |
| + Oracle-LA post-hoc | 66.09 | 36.37 | 3.23 | 36.83 | 65.94 | 26.83 | 0.83 | 32.72 | 64.17 | 18.89 | 0.00 | 29.07 |
| + Oracle-LA | 1.91 | 8.11 | 10.47 | 6.65 | 0.20 | 2.57 | 14.20 | 4.96 | 0.00 | 2.43 | 13.70 | 4.96 |
| + RECORDS | 62.91 | 37.77 | 9.17 | **37.99** | 59.20 | 36.89 | 7.10 | **35.76** | 57.29 | 34.43 | 6.97 | **34.19** |
| vs. PRODEN | -1.03 | +8.03 | +8.57 | +5.02 | -6.34 | +16.80 | +7.07 | +5.78 | -5.60 | +21.74 | +6.97 | +7.74 |
| LW | 63.69 | 22.31 | 0.00 | 30.10 | 62.94 | 10.09 | 0.00 | 25.56 | 58.31 | 6.20 | 0.00 | 22.58 |
| + Oracle-LA post-hoc | 58.60 | 25.51 | 3.07 | 29.31 | 57.89 | 9.37 | 3.40 | 24.56 | 53.40 | 6.37 | 3.06 | 21.84 |
| + Oracle-LA | 14.03 | 17.37 | 11.23 | 14.36 | 1.34 | 6.74 | 16.33 | 7.73 | 0.00 | 6.57 | 5.17 | 6.85 |
| + RECORDS | 61.41 | 24.39 | 4.45 | **31.51** | 61.54 | 10.60 | 4.33 | **26.65** | 59.02 | 9.51 | 3.26 | **25.06** |
| vs. LW | -2.28 | +2.08 | +4.45 | +1.41 | -1.40 | +0.51 | +4.33 | +1.09 | +0.71 | +3.31 | +3.26 | +2.48 |
| CAVL | 54.94 | 21.29 | 1.93 | 27.26 | 42.17 | 14.74 | 2.03 | 20.53 | 18.37 | 2.06 | 0.63 | 7.34 |
| + Oracle-LA post-hoc | 25.03 | 25.06 | 9.97 | 20.52 | 15.37 | 14.80 | 10.00 | 13.56 | 8.49 | 0.60 | 5.83 | 4.93 |
| + Oracle-LA | 2.20 | 6.63 | 12.57 | 6.86 | 0.00 | 2.77 | 13.97 | 5.16 | 0.00 | 1.20 | 14.60 | 4.80 |
| + RECORDS | 56.89 | 35.09 | 8.60 | **34.77** | 43.26 | 25.34 | 7.83 | **26.36** | 32.40 | 19.23 | 7.87 | **20.45** |
| vs. CAVL | +1.95 | +13.80 | +6.67 | +7.51 | +1.09 | +10.60 | +5.80 | +5.83 | +14.03 | +17.17 | +7.24 | +13.11 |

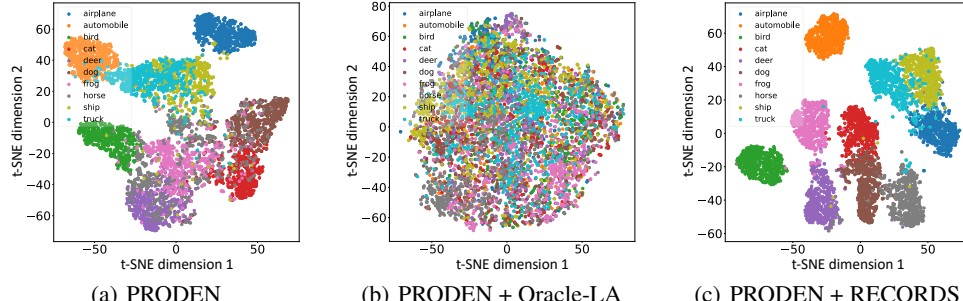

|  (a) PRODEN | (b) PRODEN + Oracle-LA | (c) PRODEN + RECORDS |
|---|---|---|

Figure 8: T-SNE of image representation on CIFAR-10-LT with $\rho = 100$ and $q = 0.5$. Note that, PRODEN + Oracle-LA post-hoc does not alter feature and shares the same figure as PRODEN.

cant improvement over the PLL baselines for varying imbalance ratio $\rho \in \{50, 100\}$ and ambiguity $q \in \{0.03, 0.05, 0.07\}$. It demonstrates the stable enhancement by our RECORDS under different potential candidate generation settings.

### E.4    FINE-GRAINED ANALYSIS ON MORE PLL BASELINES

In Table 10 and 11, we show the additional results of fine-grained analysis on CIFAR-100-LT ($\rho = 100$, $q \in \{0.03, .05, 0.07\}$) and CIFAR-100-LT-NU ($\rho = 100$, $q \in \{0.03, .05, 0.07\}$). Our RECORDS shows consistent gains on Medium and Few classes when conbined with all four PLL baselines. As a result of head-to-tail tradeoff, the accuracies of Many classes improve less or decrease slightly, which can be observed in many long-tailed learning literatures (Kang et al., 2020; Menon et al., 2021). Our RECORDS consistently improves the overall accuracies in this tradeoff.

### E.5    FEATURE VISUALIZATION

In Figure 8, We visualize the image representation produced by the feature encoder $f$ using t-SNE (van der Maaten & Hinton, 2008) on CIFAR-10-LT with $\rho = 100$ and $q = 0.5$. We can observe that there is some overlap in the representations of different classes of PRODEN due to the heavy class imbalance and the label ambiguity. PRODEN + Oracle-LA basically fails to learn discriminative representations due to the tough constant rebalancing. In contrast, our dynamic rebalancing method RECORDS produces more distinguishable representations.

Table 12: Top-1 accuracy on three benchmark datasets. Bold indicates the superior results.

| | CIFAR-10-LT | | | | | | CIFAR-100-LT | | | | | | PASCAL VOC |
|---|---|---|---|---|---|---|---|---|---|---|---|---|---|
| Imbalance ratio $\rho$ | 50 | | | 100 | | | 50 | | | 100 | | | |
| Ambiguity $q$ | 0.3 | 0.5 | 0.7 | 0.3 | 0.5 | 0.7 | 0.03 | 0.05 | 0.07 | 0.03 | 0.05 | 0.07 | |
| CORR | 76.12 | 56.45 | 41.56 | 66.38 | 50.09 | 38.11 | 42.29 | 38.03 | 36.59 | 38.39 | 34.09 | 31.05 | 24.43 |
| + Oracle-LA post-hoc | 80.70 | 58.49 | 43.44 | 72.96 | 54.64 | 41.66 | 46.94 | 40.76 | 39.07 | 41.49 | 36.79 | 33.32 | 34.12 |
| + Oracle-LA | 36.27 | 17.61 | 12.77 | 29.97 | 15.80 | 11.17 | 22.56 | 5.59 | 3.12 | 11.37 | 3.32 | 1.98 | 52.51 |
| + Oracle-BCL | 43.76 | 25.24 | 17.34 | 33.23 | 18.34 | 15.72 | 27.23 | 10.34 | 8.34 | 14.34 | 5.82 | 4.10 | 53.23 |
| + Oracle-PaCO | 42.45 | 26.34 | 16.23 | 34.21 | 18.23 | 14.23 | 27.37 | 11.23 | 7.23 | 15.82 | 5.62 | 4.72 | 53.19 |
| + LDD | 75.23 | 58.02 | 42.14 | 67.59 | 51.38 | 39.28 | 42.91 | 39.05 | 36.98 | 39.20 | 34.87 | 32.15 | 25.45 |
| + SFN | 75.98 | 57.13 | 41.77 | 66.91 | 50.23 | 38.71 | 43.14 | 38.62 | 37.03 | 39.31 | 33.91 | 31.67 | 25.63 |
| + RECORDS | **82.57** | **80.28** | **67.24** | **77.66** | **72.90** | **57.46** | **48.06** | **45.56** | **42.51** | **42.25** | **40.59** | **38.65** | **56.46** |
| vs. CORR | +6.45 | +23.83 | +25.68 | +11.28 | +22.81 | +19.35 | +5.77 | +7.53 | +5.92 | +3.86 | +6.40 | +7.60 | +32.03 |
| PRODEN | 73.12 | 54.45 | 41.37 | 63.55 | 47.37 | 38.06 | 39.23 | 35.45 | 33.90 | 34.52 | 32.04 | 29.40 | 22.39 |
| + Oracle-LA post-hoc | 77.41 | 57.14 | 42.91 | 70.71 | 48.79 | 41.38 | 43.40 | 38.64 | 35.82 | 38.40 | 35.20 | 31.92 | 31.53 |
| + Oracle-LA | 27.18 | 16.97 | 11.52 | 19.51 | 14.11 | 11.17 | 12.37 | 4.09 | 2.64 | 6.79 | 2.73 | 1.98 | 48.33 |
| + Oracle-BCL | 31.66 | 21.92 | 14.97 | 24.14 | 18.51 | 13.85 | 16.42 | 8.45 | 7.63 | 10.28 | 7.58 | 6.01 | 49.71 |
| + Oracle-PaCO | 31.48 | 21.09 | 14.58 | 23.82 | 19.00 | 15.76 | 16.34 | 8.05 | 7.06 | 11.66 | 7.52 | 7.40 | 50.04 |
| + LDD | 72.89 | 54.89 | 41.84 | 63.88 | 47.73 | 38.08 | 39.68 | 35.36 | 34.03 | 34.49 | 32.64 | 29.81 | 22.79 |
| + SFN | 73.00 | 54.73 | 41.72 | 64.16 | 47.89 | 38.02 | 39.32 | 36.04 | 34.19 | 35.13 | 32.60 | 29.51 | 22.94 |
| + RECORDS | **79.48** | **76.73** | **65.31** | **72.15** | **65.22** | **52.26** | **44.56** | **41.31** | **39.26** | **39.13** | **37.23** | **35.26** | **52.65** |
| vs. PRODEN | +6.36 | +22.28 | +23.94 | +8.60 | +17.85 | +14.2 | +5.33 | +5.86 | +5.36 | +4.61 | +5.19 | +5.86 | +30.26 |
| LW | 70.11 | 37.67 | 22.73 | 64.78 | 39.57 | 23.54 | 35.54 | 29.50 | 27.86 | 31.58 | 28.09 | 24.65 | 19.41 |
| + Oracle-LA post-hoc | 74.34 | 40.27 | 25.34 | 69.60 | 42.34 | 27.35 | 35.47 | 28.80 | 27.27 | 31.03 | 26.96 | 23.20 | 21.06 |
| + Oracle-LA | 41.90 | 21.36 | 15.28 | 25.75 | 20.35 | 14.24 | 30.37 | 14.43 | 4.79 | 30.30 | 5.08 | 2.70 | 51.53 |
| + Oracle-BCL | 45.89 | 23.38 | 18.46 | 28.77 | 23.95 | 15.51 | 33.84 | 17.75 | 8.04 | 33.55 | 8.83 | 6.18 | 52.06 |
| + Oracle-PaCO | 45.02 | 23.10 | 18.61 | 28.65 | 23.66 | 15.25 | 33.28 | 17.80 | 7.91 | 33.83 | 7.81 | 6.02 | 51.54 |
| + LDD | 70.20 | 37.87 | 22.42 | 64.90 | 39.49 | 23.44 | 35.23 | 29.05 | 27.20 | 31.00 | 28.27 | 24.07 | 19.71 |
| + SFN | 69.34 | 37.04 | 22.82 | 64.00 | 39.59 | 23.52 | 35.46 | 28.87 | 27.14 | 31.19 | 27.50 | 24.54 | 18.68 |
| + RECORDS | **76.02** | **57.39** | **40.28** | **71.18** | **57.23** | **41.24** | **36.56** | **31.67** | **29.39** | **33.00** | **28.85** | **25.64** | **53.09** |
| vs. LW | +5.91 | +19.72 | +17.55 | +6.40 | +17.66 | +17.70 | +1.02 | +2.17 | +1.53 | +1.42 | +0.76 | +0.99 | +33.68 |
| CAVL | 56.73 | 40.27 | 18.52 | 54.28 | 38.97 | 17.28 | 29.63 | 17.31 | 8.34 | 28.29 | 25.39 | 8.20 | 17.25 |
| + Oracle-LA post-hoc | 55.23 | 39.76 | 18.34 | 51.37 | 37.28 | 14.58 | 29.65 | 14.86 | 5.76 | 28.34 | 26.27 | 5.80 | 22.27 |
| + Oracle-LA | 22.16 | 14.97 | 11.50 | 18.29 | 14.23 | 10.67 | 17.31 | 4.36 | 2.83 | 7.24 | 2.55 | 2.03 | 50.78 |
| + Oracle-BCL | 24.15 | 17.82 | 12.71 | 20.02 | 15.45 | 13.10 | 19.56 | 26.96 | 5.79 | 10.70 | 3.37 | 3.68 | 51.69 |
| + Oracle-PaCO | 24.60 | 18.42 | 13.51 | 21.55 | 16.04 | 12.02 | 17.97 | 6.33 | 6.00 | 9.64 | 4.12 | 4.00 | 51.80 |
| + LDD | 56.35 | 40.28 | 18.40 | 54.33 | 39.15 | 16.49 | 29.06 | 17.26 | 8.11 | 28.19 | 25.14 | 7.60 | 16.65 |
| + SFN | 56.74 | 41.29 | 18.33 | 54.87 | 39.80 | 16.89 | 29.31 | 17.46 | 9.33 | 28.34 | 25.39 | 9.00 | 18.31 |
| + RECORDS | **67.27** | **61.23** | **40.71** | **64.35** | **58.27** | **37.38** | **42.25** | **36.53** | **29.13** | **36.93** | **31.49** | **24.98** | **53.07** |
| vs. CAVL | +10.54 | +20.96 | +22.19 | +10.07 | +19.30 | +20.1 | +12.62 | +19.22 | +14.27 | +8.64 | +6.10 | +16.78 | +35.82 |

Table 13: Results with the addition of Mixup on benchmark datasets. The best and second best results are respectively in bold and underline.

| | CIFAR-10-LT | | | | | | CIFAR-100-LT | | | | | | PASCAL VOC |
|---|---|---|---|---|---|---|---|---|---|---|---|---|---|
| Imbalance ratio $\rho$ | 50 | | | 100 | | | 50 | | | 100 | | | |
| Ambiguity $q$ | 0.3 | 0.5 | 0.7 | 0.3 | 0.5 | 0.7 | 0.03 | 0.05 | 0.07 | 0.03 | 0.05 | 0.07 | |
| LW | 71.31 | 39.69 | 25.23 | 66.08 | 41.2 | 26.59 | 37.34 | 31.59 | 30.6 | 33.78 | 30.69 | 27.91 | 20.73 |
| CAVL | 57.77 | 41.68 | 21.32 | 54.28 | 39.47 | 20.67 | 23.01 | 21.29 | 14.23 | 31.65 | 28.77 | 13.65 | 18.33 |
| PRODEN | 74.22 | 56.25 | 45.11 | 65.72 | 49.23 | 42.26 | 43.02 | 40.15 | 39.1 | 38.66 | 36.23 | 35.52 | 24.47 |
| CORR | 77.53 | 58.35 | 45.66 | 69.12 | 50.96 | 42.87 | 45.95 | 42.83 | 41.72 | 41.79 | 38.47 | 36.96 | 26.72 |
| SoLar | _83.88_ | 76.55 | 54.61 | _75.38_ | _70.63_ | 53.15 | 47.93 | _46.85_ | _45.1_ | 42.51 | _41.71_ | 39.15 | _56.49_ |
| LW+RECORDS | 77.22 | 59.88 | 43.98 | 72.2 | 59.87 | 43.63 | 38.16 | 33.6 | 31.53 | 35.33 | 31.55 | 28.8 | 55.19 |
| CAVL+RECORDS | 68.93 | 63.59 | 44.61 | 66.65 | 59.62 | 41.88 | 46.13 | 41.93 | 34.01 | 41.33 | 35.79 | 31.29 | 55.03 |
| PRODEN+RECORDS | 81.23 | _78.82_ | _68.03_ | 73.92 | 66.21 | _55.73_ | _48.12_ | 46.01 | 44.19 | _42.98_ | 41.03 | _40.25_ | 54.78 |
| CORR+RECORDS | **84.25** | **82.5** | **71.24** | **79.79** | **74.07** | **62.25** | **52.08** | **50.58** | **47.91** | **46.57** | **45.22** | **44.73** | **58.45** |

## E.6 COMPARISON WITH MORE BASELINES FOR IMBALANCED LEARNING

We additionally compare our RECORDS with two contrastive-based state-of-the-art long-tailed learning baselines, BCL (Zhu et al., 2022) and PaCO (Cui et al., 2021), and two regularization methods for mitigating imbalance in PLL, LDD (Liu et al., 2021) and SFN (Liu et al., 2021). Similar to Oracle-LA, we use the otherwise invisible oracle category distributions for BCL and PaCO, denoted as Oracle-BCL and Oracle-PaCO. We use pseudo-labels to construct positive and negative sample pairs for the contrastive learning part of BCL and PaCO, because the true labels are not visible during training. In Table 12, we show the top-1 accuracy on CIFAR-10-LT and CIFAR-100-LT under the best PLL baseline CORR. Both BCL and PaCO use an LA-like form for the supervised part. Despite the performance gains from contrastive learning (compared to Oracle-LA), they also experience performance degradation similar to that of LA. LDD and SFN implicitly mitigate the imbalance in PLL by regularization constraints on the parameter space, and achieve a slight improvement over the PLL baseline CORR. However, they do not explicitly address the long-tailed distribution problem of LT-PLL. Our method achieves a significant improvement over all baselines. We should also note that, in these baselines except LDD and SFN, they use the oracle class distribution, which our method also does not use. Therefore, our method is more general than them.

### E.7    COMPARISON WITH THE CONCURRENT LT-PLL WORK SOLAR

SoLar (Wang et al., 2022a) is a concurrent LT-PLL work published in NeuIPS 2022. At the time of our submission, we did not have access to this work and make comparisons with it. It improves the label disambiguation process in LT-PLL through the optimal transport technique. However, it requires an extra outer-loop to refine the label prediction via sinkhorn-knopp iteration, which increases the computational complexity. Different from SoLar, this paper tries to solve the LT-PLL problem from the perspective of rebalancing in a lightweight and effective manner.

Note that compared to other PLL baselines and previous experiments in this paper, SoLar additionally applies Mixup (Zhang et al., 2018) in the experiments. To align the experimental setup, we show the results with the addition of Mixup in Table 13. All other settings remain the same as previous sections. Compared to SoLar, which designed for the LT-PLL problem, our method still offers consistent improvements. The code implementation of comparisons with Solar can be found here.

