# OpenReview forum: "Long-Tailed Partial Label Learning via Dynamic Rebalancing"
_ICLR.cc/2023/Conference — ICLR 2023 poster_

### Official Review · Reviewer_MogU · 2022-10-25

**Confidence:** 4
**Clarity, Quality, Novelty And Reproducibility:** 1. Despite the good results, this pap…
**Correctness:** 3
**Technical Novelty And Significance:** 3
**Empirical Novelty And Significance:** Not applicable
**Recommendation:** 6

**Strength And Weaknesses:**

Strength:
1. Well-motivated. The study of complex label distributions of data is a hot research area. This paper introduces a more difficult but common LT_PLL scenario by combining long-tailed and partial label learning.
2. Well-written. This paper is clearly organized and easy to follow.
3. The proposed method is reasonable and effective, as shown in experimental results.

Weakness:
1. There are other work focusing on similar scenarios, e.g. [1], so this paper is not the first attempt. It would be better to include some discussion and experimental comparison.
2. There are some grammars. For example, in the last paragraph of Page 1 , “In Figure 1, we trace the average prediction of a PLL model PRODEN Lv et al. (2020) “, \citep should be used here, i.e., “PRODEN Lv et al. (2020)” should be PRODEN (Lv et al. , 2020). There are many such mis-uses of \cite in this paper.
“Works” should be replaced by studies
3. Please explain how the approximation in Equation (4) is obtained and what is the gap between these two items. Also explain how the second equation is obtained.
4. The toy data are not a good testbed for the long-tailed and partial labeling setting, as there are no partial labels there.

Ref:
[1] Liu W, Wang L, Chen J, et al. A Partial Label Metric Learning Algorithm for Class Imbalanced Data[C]//Asian Conference on Machine Learning. PMLR, 2021: 1413-1428.


**Summary Of The Paper:**

This paper focuses on a more challenging but practically real-world data scenario, which couples the label ambiguity and heavy imbalance. This paper first demonstrates the infeasibility of the straightforward combination of the current long-tailed learning and partial label learning. Then it proposes a dynamic rebalancing method, termed RECORDS, to rebalance the training process. Experimental results on several benchmark datasets show the significant gain of RECORDS compared with a range of baselines.

**Summary Of The Review:**

The proposed method is well-motivated and seems to be effective, and experimental results and analysis are quite adequate. However, The experimental evaluation requires improvement, and code should be submitted along with the paper or made publicly available.

---

> ### Author Response · Authors · 2022-11-14
> **Response to Reviewer MogU[2/2]**
>
> > **Q3:** Please explain how the approximation in Equation (4) is obtained and what is the gap between these two items. Also explain how the second equation is obtained.
>
> **A3:** The approximation is from the idea of normalized weighted geometric mean (NWGM) [1], which has been studied and used in some previous works, e.g., Equation (51) in [2], Section 4.2 in [3] and Equation (4) in [4]. The specific explanation of this approximation can be found in Section 4 and Section 5 in [2]. It helps us capture more stable feature statistics and combine with the latest updated linear classifier to estimate the surrogate class distribution. We will revise the submission by adding more citations and discussion into the corresponding paragraphs to clarify this operator.
>
> Regarding the second equation, it holds the equality because $z = g\circ f$ and $g$ is a linear layer (please see Section 3.1 about its definition), and thus we can exchange the outside term as the inside input with holding out the parameter $W$.
>
> [1] Baldi, P., \& Sadowski, P. J. Understanding dropout. Advances in neural information processing systems, 2013.
>
> [2] Baldi, P., \& Sadowski, P. The dropout learning algorithm. Artificial intelligence, 2014.
>
> [3] Xu, K., Ba, J., Kiros, R., Cho, K., Courville, A., Salakhudinov, R., ... \& Bengio, Y. Show, attend and tell: Neural image caption generation with visual attention. In International conference on machine learning, 2015.
>
> [4] Wang, T., Huang, J., Zhang, H., \& Sun, Q. Visual commonsense r-cnn. In Proceedings of the IEEE/CVF Conference on Computer Vision and Pattern Recognition, 2020.
>
> > **Q4:** The toy data are not a good testbed for the long-tailed and partial labeling setting, as there are no partial labels there.
>
> **A4:** Actually, as described in Section 5.1---"The label set of each sample consists of a true label and negative labels having a 0.6 probability flipped from other classes.", we generated candidate label sets for the toy dataset and used the candidate set during training. We will highlight the corresponding contents to avoid misunderstanding.
>
> > **Q5:** Despite the good results, this paper is incremental compared to the LA.
>
> **A4:** We would like to kindly defend our proposal. We would like to point out three points: 1) LA and its recent state-of-the-art variants like  BCL [1] and PaCO [2] requires the oracle class distribution to rebalance the training, while our method does not require this; 2) The reason that why LA still performs poorly in LT-PLL, even worse than the vanilla PLL methods, has not been discovered in previous works. We analyze the LT-PLL problem in depth and first point out that the key challenge lies in the mismatch between the training dynamics and constant rebalancing; 3) We compare a range of methods including new baselines proposed by other reviewers, and the extensive results demonstrate the advantages of our method by a significant and consistent improvement. Besides, we present a theoretical proof on how our method connects with LA when the small ambiguity degree condition satisfies, which maintains a desired methodology characteristic.
>
> [1] Zhu, J., Wang, Z., Chen, J., Chen, Y. P. P., \& Jiang, Y. G. Balanced Contrastive Learning for Long-Tailed Visual Recognition. In Proceedings of the IEEE/CVF Conference on Computer Vision and Pattern Recognition, 2022.
>
> [2] Cui, J., Zhong, Z., Liu, S., Yu, B., \& Jia, J. Parametric contrastive learning. In Proceedings of the IEEE/CVF international conference on computer vision, 2021.
>
> > **Q6:** No transparency and reproducibility. The authors have not submitted their code, which is a red light to me. It's 2022 and submitting the code along with the paper should be the norm.
>
> **A6:** Thank you for pointing out this. To avoid the concerns on the reproduciabitliy and the detailed setups, we will open our source code in an anonymous repository.

---

> ### Author Response · Authors · 2022-11-14
> **Response to Reviewer MogU[1/2]**
>
> Thank you for your time devoted to reviewing this paper and your constructive suggestions. Here are our detailed replies to your questions.
>
> > **Q1:** There are other work focusing on similar scenarios, e.g. [1], so this paper is not the first attempt. It would be better to include some discussion and experimental comparison.
>
> **A1:** Thanks for recommending this work [1]. We have modified the statement in the paper and included [1] into discussion. Specifically, [1] studied partial label metric learning algorithms for class imbalanced data, and proposed two regularization terms to induce class uniformity. As the original formulation cannot directly apply into deep-learning-based methods, we adapt it to our deep partial label learning baselines by incorporating their two regularization terms, respectively termed as LDD and SFN. Besides, we also add two state-of-the-art variants of logit adjustment namely, BCL [2] and PaCO [3], as our baselines. The results are summarized as follows.
>
> [**Table 1.** Comparison with different baselines on CIFAR-10-LT]
> | Imbalance ratio $\rho$ |  50   |  50   |  50   |  100  |  100  |  100  |
> | ---------------------- |:-----:|:-----:|:-----:|:-----:|:-----:|:-----:|
> | ambiguity $q$          |  0.3  |  0.5  |  0.7  |  0.3  |  0.5  |  0.7  |
> | CORR                   | 76.12 | 56.45 | 41.56 | 66.38 | 50.09 | 38.11 |
> | + Oracle-LA            | 36.27 | 17.61 | 12.77 | 29.97 | 15.80 | 11.75 |
> | + Oracle-LA-Posthoc    | 80.70 | 58.49 | 43.44 | 72.96 | 54.64 | 41.66 |
> | + Oracle-BCL           | 43.76 | 25.24 | 17.34 | 33.23 | 18.34 | 15.72 |
> | + Oracle-PaCO          | 42.45 | 26.34 | 16.23 | 34.21 | 18.23 | 14.23 |
> | + LDD                  | 75.23 | 58.02 | 42.14 | 67.59 | 51.38 | 39.28 |
> | + SFN                  | 75.98 | 57.13 | 41.77 | 66.91 | 50.23 | 38.71 |
> | + RECORDS              | **82.57** | **80.28** | **67.24** | **77.66** | **72.90** | **57.46** |
>
> [**Table 2.** Comparison with different baselines on CIFAR-100-LT]
> | Imbalance ratio $\rho$ |     50     |     50     |     50     |     100    |     100    |     100    |
> |------------------------|:----------:|:----------:|:----------:|:----------:|:----------:|:----------:|
> | ambiguity $q$          | 0.03       | 0.05       | 0.07       | 0.03       | 0.05       | 0.07       |
> | CORR                   | 42.29      | 38.03      | 36.59      | 38.39      | 34.09      | 31.05      |
> | + Oracle-LA            | 22.56      | 5.59       | 3.12       | 11.37      | 3.32       | 1.98       |
> | + Oracle-LA-Posthoc    | 46.94      | 40.76      | 39.07      | 41.49      | 36.79      | 33.32      |
> | + Oracle-BCL           | 27.23      | 10.34      | 8.34       | 14.34      | 5.82       | 4.10       |
> | + Oracle-PaCO          | 27.37      | 11.23      | 7.23       | 15.82      | 5.62       | 4.72       |
> | + LDD                  | 42.91      | 39.05      | 36.98      | 39.20      | 34.87      | 32.15      |
> | + SFN                  | 43.14      | 38.62      | 37.03      | 39.31      | 33.91      | 31.67      |
> | + RECORDS| **48.06** | **45.56** | **42.51** | **42.25** | **40.59** | **38.65** |
>
> From the results, we can find see SFN and LDD slightly improve the performance of CORR and are better than Oracle-LA and its variants Oracle-BCL, Oracle-PaCO. However, RECORDS still significantly outperforms these methods, showing the effectiveness of dynamic rebalancing.
>
>
> [1] Liu, W., Wang, L., Chen, J., Zhou, Y., Zheng, R., \& He, J. A Partial Label Metric Learning Algorithm for Class Imbalanced Data. In Asian Conference on Machine Learning, 2021.
>
> [2] Zhu, J., Wang, Z., Chen, J., Chen, Y. P. P., \& Jiang, Y. G. Balanced Contrastive Learning for Long-Tailed Visual Recognition. In Proceedings of the IEEE/CVF Conference on Computer Vision and Pattern Recognition, 2022.
>
> [3] Cui, J., Zhong, Z., Liu, S., Yu, B., \& Jia, J. Parametric contrastive learning. In Proceedings of the IEEE/CVF international conference on computer vision, 2021.
>
> > **Q2:** There are some grammars. For example, in the last paragraph of Page 1 , “In Figure 1, we trace the average prediction of a PLL model PRODEN Lv et al. (2020) “, \citep should be used here, i.e., “PRODEN Lv et al. (2020)” should be PRODEN (Lv et al. , 2020). There are many such mis-uses of \cite in this paper. “Works” should be replaced by studies
>
> **A2:** Thanks for pointing them out! We have corrected them and proofread the submission again.

---

> ### Author Response · Authors · 2022-11-17
> **Kind Reminder to Respond Our Rebuttal**
>
> Dear Reviewer MogU,
>
> We would like to thank you again for your efforts and time in providing constructive feedback and comments. We have carefully consider your advices/questions and have provided as much refinement and experiments to address your concerns regarding the submission. Since there are now only **one day** remaining that we are allowed to further modify the submission. Would you mind checking the current **updated submission** and confirming whether we have solved your concerns. If there are any further questions or suggestions, we would like to conduct the experiments or refine the description to improve our submission.
>
> Sincerely
>
> The authors

---

> ### Author Response · Authors · 2022-11-23
> **Looking forward to your responses or further suggestions/comments!**
>
> Dear Reviewer MogU,
>
> We have carefully considered and addressed your initial concerns regarding our paper. We are happy to discuss them with you in the openreview system if you feel that there still are some concerns/questions. We also welcome new suggestions/comments from you!
>
> Best regards,
>
> The authors of Paper3732

---

> ### Author Response · Authors · 2022-11-26
> **We anticipate your feedback!**
>
> Dear Reviewer MogU,
>
> The authors greatly appreciate your time and effort in reviewing this submission, and eagerly await your response.
>
> We have provided detailed responses to every one of your concerns/questions. Please help us to review our responses once again and kindly let us know whether they fully or partially address your concerns and if our explanations are in the right direction.
>
> Best,
>
> The authors of Paper3732

---

> ### Author Response · Authors · 2022-12-05
> **We are anticipating your post-rebuttal feedback!**
>
> Dear Reviewer MogU,
>
> We would like to sincerely thank you again for your time in reviewing our work!
>
> We understand you might be quite busy. However, as the discussion deadline is approaching, would you mind checking our response and confirming whether you have any further questions? Any further comments and discussions are welcomed!
>
> Best Regards,
>
> The authors of Paper3732

---

### Official Review · Reviewer_dMZw · 2022-10-26

**Confidence:** 4
**Correctness:** 4
**Technical Novelty And Significance:** 4
**Empirical Novelty And Significance:** 4
**Recommendation:** 8

**Clarity, Quality, Novelty And Reproducibility:**

The paper is generally clear, especially in the essential parts, although some issues with Theorem 1 were noted above. The intuition-building and simple examples are particularly compelling in the presentation. To my knowledge this seems to be new, although I am not particularly steeped in this corner of the literature. Code is not provided with the submission, but experiments seem straightforwardly reproducible.

**Strength And Weaknesses:**

Strengths:
 + The paper does a nice job setting up the LT + PLL problem, and showing that naive combinations of existing methods tend to fail.
 + Each of the LT and PLL problems are well-motivated, and prior work seems to be well-cited.
 + The method is explained well and empirical results are compelling, with appropriate ablations and comparisons.
 + The authors construct a new benchmark dataset, CIFAR-100-LT-NU

Weaknesses (and questions):
- It is still somewhat mysterious why using the oracle class probabilities fails so badly with PLL methods. Is the main pathology that the model simply doesn't train well at all? E.g., if one had an identically distributed test set with the same class distribution as training, would PLL methods using constant rebalancing behave poorly there too? Would "tempering" in the oracle logit adjustment (i.e., temperature scaling P(y) so that you start close to uniform and eventually adjusting for P(y)) lead to a similar result as the dynamic rebalancing? This question doesn't undercut the method in the paper (since it requires no access to the oracle), but it would be useful to shed a bit more light on how the oracle LA is disrupting training dynamics. The linear probing analysis does some of this.
- Most of Theorem 1 seems to be a restatement of results from Liu and Dietterich. Specifying exactly what is added here would be useful. For example, it seems like one could assume that the Liu and Dietterich error bound holds, and Theorem 1 would be a relatively straightforward corollary (Note: the straightforwardness would not be a weakness!). Perhaps this is wrong, but some explicit discussion would be useful to understand what is novel.
 - Switching the softmax and expectation in (4) seems questionable as an approximation, especially in the long-tailed context where we expect some rare or common class logits to lie outside of the region where the softmax is approximately linear. The justification given in the text could at least use a citation if not some elaboration (say in an appendix). Alternatively, could this be formulated as a form of stabilization or regularization via Jensen's inequality?

**Summary Of The Paper:**

The authors consider a problem where data exhibit both a long-tail (LT), where there is major class imbalance that we do not expect at test time, and PLL (partially label learning), where there exists a true label but at training time we only observe a superset that contains the true label. The authors show that standard LT adjustment interact poorly with PLL, even if the oracle class distribution is known. The authors propose a solution that applies a dynamic LT adjustment that is more consistent with the model's current predictions at each step of training. This dynamic adjustment can be used with any PLL algorithm. The authors show that this scheme will lead to an LT adjustment that converges to the oracle, but argue that the dynamic nature of the adjustment allows the model to train better. This last claim is substantiated with several empirical experiments that consider how different LT rebalancing schemes compare with the same PLL algorithm, as well as some studies exploring and probing representations, and a study examining the effect of the momentum tuning parameter that controls the dynamics of the class adjustment.

**Summary Of The Review:**

This is a nice, thorough treatment of this problem, with a method that appears to be quite successful. The authors do a nice job of motivating their work, and showing how their method fits modularly with current PLL methods. The experimental comparisons are generally compelling. The theory could stand to be better-modularized, but that is a minor concern. The paper could potentially be improved by showcasing a "killer app" where the LT-PLL problem arises.

---

> ### Author Response · Authors · 2022-11-14
> **Response to Reviewer dMZw[2/2]**
>
> > **Q2:** Most of Theorem 1 seems to be a restatement of results from Liu and Dietterich. Specifying exactly what is added here would be useful. For example, it seems like one could assume that the Liu and Dietterich error bound holds, and Theorem 1 would be a relatively straightforward corollary (Note: the straightforwardness would not be a weakness!). Perhaps this is wrong, but some explicit discussion would be useful to understand what is novel.
>
> **A2:** Thank you very much for the advice. Our theorem 1 is indeed built upon the theory in [1], and we will add the clear emphasis in the corresponding part to clarify this point. The necessity of Theorem 1 in our submission is mainly to study the training limit of our dynamic rebalancing method when the small ambiguity degree condition satisfies. Roughly speaking, if our method can degenerate to the constant LA in this limit, it seems to be nicely aligned with the classical long-tailed learning methodology with the true labels. As shown in the theoretical analysis and the empirical verification in Figure 3, this characteristic is fortunately maintained in our method.
>
> However, we do agree with the reviewer's suggestion to avoid the unnecessary decoration in the theoretical analysis and will clearly clarify which part is from the previous study.
>
> [1] Liu, L., \& Dietterich, T. Learnability of the superset label learning problem. In International Conference on Machine Learning (pp. 1629-1637), 2014.
>
> > **Q3:** Switching the softmax and expectation in (4) seems questionable as an approximation, especially in the long-tailed context where we expect some rare or common class logits to lie outside of the region where the softmax is approximately linear. The justification given in the text could at least use a citation if not some elaboration (say in an appendix). Alternatively, could this be formulated as a form of stabilization or regularization via Jensen's inequality?
>
> **A3:** We appreciate the reviewer's suggestion and will add the necessary citations about this approximation into the submission. Formally, it is from the idea of normalized weighted geometric mean (NWGM) [1], which has been studied and used in some previous works, e.g., Equation (51) in [2], Section 4.2 in [3] and Equation (4) in [4]. This approximation helps us capture more stable feature statistics and combine with the latest updated linear classifier to estimate the surrogate class distribution.
>
> [1] Baldi, P., \& Sadowski, P. J. Understanding dropout. Advances in neural information processing systems, 2013.
>
> [2] Baldi, P., \& Sadowski, P. The dropout learning algorithm. Artificial intelligence, 2014.
>
> [3] Xu, K., Ba, J., Kiros, R., Cho, K., Courville, A., Salakhudinov, R., ... \& Bengio, Y. Show, attend and tell: Neural image caption generation with visual attention. In International conference on machine learning, 2015.
>
> [4] Wang, T., Huang, J., Zhang, H., \& Sun, Q. Visual commonsense r-cnn. In Proceedings of the IEEE/CVF Conference on Computer Vision and Pattern Recognition, 2020.
>
> > **Q4:** The paper could potentially be improved by showcasing a "killer app" where the LT-PLL problem arises.
>
> **A4:** Thanks for the advice. We have added one section in the appendix about the motivating examples to support the practical intuition of LT-PLL, and refine the introduction parts.

---

> ### Author Response · Authors · 2022-11-14
> **Response to Reviewer dMZw[1/2]**
>
> Thank you for your time devoted to reviewing this paper and your constructive suggestions. Here are our detailed replies to your questions.
>
> > **Q1:** It is still somewhat mysterious why using the oracle class probabilities fails so badly with PLL methods. Is the main pathology that the model simply doesn't train well at all? E.g., if one had an identically distributed test set with the same class distribution as training, would PLL methods using constant rebalancing behave poorly there too? Would "tempering" in the oracle logit adjustment (i.e., temperature scaling P(y) so that you start close to uniform and eventually adjusting for P(y)) lead to a similar result as the dynamic rebalancing? This question doesn't undercut the method in the paper (since it requires no access to the oracle), but it would be useful to shed a bit more light on how the oracle LA is disrupting training dynamics. The linear probing analysis does some of this.
>
> **A1:** Thank you for the suggestion. If both training set and test set following the same long-tailed distribution, LA might not be a good choice, since the original intuition of LA is for the distribution shift. As can be seen in the subfigure of the 1st row and 3rd column in Figure 4, the training under LA does not well capture the true decision boundary of training set. If the test set also follows the same distribution as the training set, the performance should be similarly worse. The reason behind this phenomenon is possibly because LA is too hard for the dynamics of PLL methods, and the corresponding constant adjustment in the logit space might be too aggressive.
>
> Following the reviewer's constructive suggestion, we implement the experiments by tempering the logit adjustment (marked by "Temp Oracle-LA") and summarize the results in the below table. The results indicate: 1) Temp Oracle-LA does show effectiveness in improving the performance of Oracle-LA, which confirms the advantages of dynamic rebalancing. 2) RECORDS still consistently outperforms Temp Oracle-LA in the performance, and shows an increasing gain along with the increasing ambiguity degree. This is mainly because the tempering strategy cannot well align the training dynamic of PLL, while our method directly estimates the surrogate class distribution from the feature space constructed by PLL methods better matches their training dynamics.
>
> We appreciate the reviewer's questions about the possible ablation of this problem, and will include these experimental results into the draft to facilitate the potential understanding.
>
> [**Table 1.** Comparison with different baselines on CIFAR-10-LT]
> | Imbalance ratio $\rho$ | 50        | 50        | 50        | 50        | 50        | 100       |
> | ---------------------- | --------- | --------- | --------- | --------- | --------- | --------- |
> | Ambiguity $q$          | 0.3       | 0.5       | 0.7       | 0.3       | 0.5       | 0.7       |
> | CORR                   | 76.12     | 56.45     | 41.56     | 66.38     | 50.09     | 38.11     |
> | + Oracle-LA            | 36.27     | 17.61     | 12.77     | 29.97     | 15.80     | 11.75     |
> | + Oracle-LA-Posthoc    | 80.70     | 58.49     | 43.44     | 72.96     | 54.64     | 41.66     |
> | + Temp Oracle-LA       | 81.37     | 43.62     | 18.10     | 76.09     | 25.88     | 16.11     |
> | + RECORDS              | **82.57** | **80.28** | **67.24** | **77.66** | **72.90** | **57.46** |
>
> [**Table 2.** Comparison with different baselines on CIFAR-100-LT]
> | Imbalance ratio $\rho$   | 50         | 50         | 50         | 50         | 50         | 100        |
> |--------------------------|------------|------------|------------|------------|------------|------------|
> | Ambiguity $q$            |    0.03    |    0.05    |    0.07    |    0.03    |    0.05    |    0.07    |
> | CORR                     |   42.29    |   38.03    |   36.59    |   38.39    |   34.09    |   31.05    |
> | + Oracle-LA              |   22.56    |    5.59    |    3.12    |   11.37    |    3.32    |    1.98    |
> | + Oracle-LA-Posthoc      |   46.94    |   40.76    |   39.07    |   41.49    |   36.79    |   33.32    |
> | + Temp Oracle-LA  |   47.44    |   43.46    |   29.75    |   41.78    |   39.19    |   33.69    |
> | + RECORDS                | **48.06** | **45.56** | **42.51** | **42.25** | **40.59** | **38.65** |

---

> ### Author Response · Authors · 2022-11-17
> **Kind Reminder to Respond Our Rebuttal**
>
> Dear Reviewer dMZw,
>
> We would like to thank you again for your efforts and time in providing constructive feedback and comments. We have carefully consider your advices/questions and have provided as much refinement and experiments to address your concerns regarding the submission. Since there are now only **one day** remaining that we are allowed to further modify the submission. Would you mind checking the current **updated submission** and confirming whether we have solved your concerns. If there are any further questions or suggestions, we would like to conduct the experiments or refine the description to improve our submission.
>
> Sincerely
>
> The authors

---

> ### Author Response · Authors · 2022-11-23
> **Looking forward to your responses or further suggestions/comments!**
>
> Dear Reviewer dMZw,
>
> We have carefully considered and addressed your initial concerns regarding our paper. We are happy to discuss them with you in the openreview system if you feel that there still are some concerns/questions. We also welcome new suggestions/comments from you!
>
> Best regards,
>
> The authors of Paper3732

---

> ### Author Response · Authors · 2022-11-26
> **We anticipate your feedback!**
>
> Dear Reviewer dMZw,
>
> The authors greatly appreciate your time and effort in reviewing this submission, and eagerly await your response.
>
> We have provided detailed responses to every one of your concerns/questions. Please help us to review our responses once again and kindly let us know whether they fully or partially address your concerns and if our explanations are in the right direction.
>
> Best,
>
> The authors of Paper3732

---

> ### Author Response · Authors · 2022-12-05
> **We are anticipating your post-rebuttal feedback!**
>
> Dear Reviewer dMZw,
>
> We would like to sincerely thank you again for your time in reviewing our work!
>
> We understand you might be quite busy. However, as the discussion deadline is approaching, would you mind checking our response and confirming whether you have any further questions? Any further comments are discussions are welcomed!
>
> Best Regards,
>
> The authors of Paper3732

---

> > ### Comment · Reviewer_dMZw · 2022-12-05
> > **Apologies for late reply**
> >
> > I appreciate the responses here, especially the tempering baseline. I don't have any further questions, and continue to support accepting the paper.

---

> > > ### Author Response · Authors · 2022-12-06
> > > **Thanks for your feedback and positive support**
> > >
> > >
> > > Dear Reviewer,
> > >
> > >     We do appreciate your feedback about our effort for your proposed concerns, and thanks very much for your positive support. We will add all your advice and subsequent improvement into the final version.
> > >
> > > Best
> > >
> > > The authors of Submission 3732

---

### Official Review · Reviewer_djhm · 2022-11-01

**Confidence:** 2
**Correctness:** 4
**Technical Novelty And Significance:** 2
**Empirical Novelty And Significance:** 2
**Recommendation:** 5

**Clarity, Quality, Novelty And Reproducibility:**

The paper is relatively clearly written.

The problem setting is novel, although the method is mostly using existing techniques.

The paper says that code will be available, but it is not currently available yet.

**Details Of Ethics Concerns:**

No ethics concerns

**Strength And Weaknesses:**

Strengths

 - The motivation behind the setting and the problems with existing solutions are described well.
 - Dynamic rebalancing is clearly explained and the toy example is a clear example of where it works well.
 - Dynamic rebalancing can be applied to many methods as a drop-in replacement.

Weaknesses

 - The baselines are primarily focused on long-tailed learning, so it is natural that the baselines would not handle the partial labels properly.
 - Dynamic rebalancing is a relatively obvious solution (possibly the only simpler solution is to just use the previous epoch to estimate the class distribution, and it's not clear that momentum to average would outperform this).

**Summary Of The Paper:**

This paper studies the combination of long-tailed learning and partial label learning.

Applying existing methods from long-tailed learning to this regime result in poor performance because of ambiguous labels. This paper proposes dynamic rebalancing as a solution.

**Summary Of The Review:**

The paper combines the long-tailed learning and partial label learning settings. Dynamic rebalancing improves upon the baselines, but mostly combines existing techniques together.

---

> ### Author Response · Authors · 2022-11-14
> **Response to Reviewer djhm[2/2]**
>
> > **Q2**: Dynamic rebalancing is a relatively obvious solution (possibly the only simpler solution is to just use the previous epoch to estimate the class distribution, and it's not clear that momentum to average would outperform this).
>
> **A2:** First, we would like to kindly defend our proposal, since none of rebalancing methods including some variants of logit adjustment follow the same spirit. The key uncertainty is: *why we can share the same parameter space with PLL to estimate the parametric class distribution. Will not it collapse with the PLL training?* Specially, we conduct a range of explorations and finally find such a simple yet effective design theoretically couples with the PLL training to approach the oracle logit adjustment in the limit, and empirically matches the dynamic of PLL during training to effectively improve the performance. Second, to alleviate the concern of the reviewer, we follow the reviewer's suggestion to validate some simpler variants like RECORDS without "momentum" (marked as "Epoch RECORDS") or apply the straightforward Oracle-LA in a benign manner (marked as "Temp Oracle-LA"). Besides, we also include some state-of-the-art variants with logit adjustment in long-tailed learning, "Oracle BCL"[1] and "Oracle-PaCO"[2]. The results are summarized in the following.
>
> [**Table 3.** Comparison with different baselines on CIFAR-10-LT]
> | Imbalance ratio $\rho$ | 50        | 50        | 50        | 50        | 50        | 100       |
> | ---------------------- | --------- | --------- | --------- | --------- | --------- | --------- |
> | Ambiguity $q$          | 0.3       | 0.5       | 0.7       | 0.3       | 0.5       | 0.7       |
> | CORR                   | 76.12     | 56.45     | 41.56     | 66.38     | 50.09     | 38.11     |
> | + Oracle-LA            | 36.27     | 17.61     | 12.77     | 29.97     | 15.80     | 11.75     |
> | + Oracle-LA-Posthoc    | 80.70     | 58.49     | 43.44     | 72.96     | 54.64     | 41.66     |
> | + Oracle-BCL           | 43.76 | 25.24 | 17.34 | 33.23 | 18.34 | 15.72 |
> | + Oracle-PaCO          | 42.45 | 26.34 | 16.23 | 34.21 | 18.23 | 14.23 |
> | + Temp Oracle-LA       | 81.37     | 43.62     | 18.10     | 76.09     | 25.88     | 16.11     |
> | + Epoch RECORDS        | 75.43     | 70.27     | 59.50     | 69.38     | 63.12     | 47.85     |
> | + RECORDS              | **82.57** | **80.28** | **67.24** | **77.66** | **72.90** | **57.46** |
>
> [**Table 4.** Comparison with different baselines on CIFAR-100-LT]
> | Imbalance ratio $\rho$   | 50         | 50         | 50         | 50         | 50         | 100        |
> |--------------------------|------------|------------|------------|------------|------------|------------|
> | Ambiguity $q$            |    0.03    |    0.05    |    0.07    |    0.03    |    0.05    |    0.07    |
> | CORR                     |   42.29    |   38.03    |   36.59    |   38.39    |   34.09    |   31.05    |
> | + Oracle-LA              |   22.56    |    5.59    |    3.12    |   11.37    |    3.32    |    1.98    |
> | + Oracle-LA-Posthoc      |   46.94    |   40.76    |   39.07    |   41.49    |   36.79    |   33.32    |
> | + Oracle-BCL           | 27.23      | 10.34      | 8.34       | 14.34      | 5.82       | 4.10       |
> | + Oracle-PaCO          | 27.37      | 11.23      | 7.23       | 15.82      | 5.62       | 4.72       |
> | + Temp Oracle-LA  |   47.44    |   43.46    |   29.75    |   41.78    |   39.19    |   33.69    |
> | + Epoch RECORDS     |   46.54    |   43.07    |   38.28    |   41.58    |   37.14    |   34.38    |
> | + RECORDS                | **48.06** | **45.56** | **42.51** | **42.25** | **40.59** | **38.65** |
>
> According to the results, we can find that although some variants with logit adjustments over their vanilla counterparts and some simper solutions might be effective, our methods consistently and significantly outperform them, which confirms the effectiveness of our design.
>
> However, we do appreciate the reviewer's challenge or concern about our method. We will include these results into the regular parts or appendix with the necessary discussion to improve the content.
>
> [1] Zhu, J., Wang, Z., Chen, J., Chen, Y. P. P., \& Jiang, Y. G. Balanced Contrastive Learning for Long-Tailed Visual Recognition. In Proceedings of the IEEE/CVF Conference on Computer Vision and Pattern Recognition, 2022.
>
> [2] Cui, J., Zhong, Z., Liu, S., Yu, B., \& Jia, J. Parametric contrastive learning. In Proceedings of the IEEE/CVF international conference on computer vision, 2021.
>
> > **Q3**: The paper says that code will be available, but it is not currently available yet.
>
> **A3:** Thank you for pointing out this. To avoid the concerns on the reproduciabitliy and the detailed setups, we will open our source code in an anonymous repository.

---

> ### Author Response · Authors · 2022-11-14
> **Response to Reviewer djhm[1/2]**
>
> Thank you for your time devoted to reviewing this paper and your constructive suggestions. Here are our detailed replies to your questions.
>
> > **Q1**: The baselines are primarily focused on long-tailed learning, so it is natural that the baselines would not handle the partial labels properly.
>
> **A1:** We would like to kindly clarify that all these baselines are implemented under the framework of recent state-of-the-art partial label learning methods that can intrinsically handle partial labels. However, the key problem to long-tailed learning methods (e.g., logit adjustment) is that they lack of consideration the dynamic of PLL and thus is not benign to label disambiguation in PLL.
>
> To alleviate the concern of the reviewer about the baselines, we implement an experiment to compare RECORDS with [1] recommended by Reviewer MogU. Specifically, [1] studied partial label metric learning algorithms for class imbalanced data, and proposed two regularization terms to induce class uniformity. As the original formulation cannot directly apply into deep-learning-based methods, we adapt it to our deep partial label learning baselines by incorporating their two regularization terms, respectively termed as LDD and SFN. According to the results in the below tables, we can see that our method significantly outperforms the competitive LDD and SFN in different imbalance ratios and different ambiguity degrees. We will add more comprehensive comparisons with LDD and SFN into our submission to make our results more convincing.
>
> [**Table 1.** Comparison with different baselines on CIFAR-10-LT]
> | Imbalance ratio $\rho$ |  50   |  50   |  50   |  100  |  100  |  100  |
> | ---------------------- |:-----:|:-----:|:-----:|:-----:|:-----:|:-----:|
> | ambiguity $q$          |  0.3  |  0.5  |  0.7  |  0.3  |  0.5  |  0.7  |
> | CORR                   | 76.12 | 56.45 | 41.56 | 66.38 | 50.09 | 38.11 |
> | + Oracle-LA            | 36.27 | 17.61 | 12.77 | 29.97 | 15.80 | 11.75 |
> | + Oracle-LA-Posthoc    | 80.70 | 58.49 | 43.44 | 72.96 | 54.64 | 41.66 |
> | + LDD                  | 75.23 | 58.02 | 42.14 | 67.59 | 51.38 | 39.28 |
> | + SFN                  | 75.98 | 57.13 | 41.77 | 66.91 | 50.23 | 38.71 |
> | + RECORDS              | **82.57** | **80.28** | **67.24** | **77.66** | **72.90** | **57.46** |
>
> [**Table 2.** Comparison with different baselines on CIFAR-100-LT]
> | Imbalance ratio $\rho$ |     50     |     50     |     50     |     100    |     100    |     100    |
> |------------------------|:----------:|:----------:|:----------:|:----------:|:----------:|:----------:|
> | ambiguity $q$          | 0.03       | 0.05       | 0.07       | 0.03       | 0.05       | 0.07       |
> | CORR                   | 42.29      | 38.03      | 36.59      | 38.39      | 34.09      | 31.05      |
> | + Oracle-LA            | 22.56      | 5.59       | 3.12       | 11.37      | 3.32       | 1.98       |
> | + Oracle-LA-Posthoc    | 46.94      | 40.76      | 39.07      | 41.49      | 36.79      | 33.32      |
> | + LDD                  | 42.91      | 39.05      | 36.98      | 39.20      | 34.87      | 32.15      |
> | + SFN                  | 43.14      | 38.62      | 37.03      | 39.31      | 33.91      | 31.67      |
> | + RECORDS| **48.06** | **45.56** | **42.51** | **42.25** | **40.59** | **38.65** |
>
> [1] Liu, W., Wang, L., Chen, J., Zhou, Y., Zheng, R., \& He, J. A Partial Label Metric Learning Algorithm for Class Imbalanced Data. In Asian Conference on Machine Learning, 2021.

---

> ### Author Response · Authors · 2022-11-17
> **Kind Reminder to Respond Our Rebuttal**
>
> Dear Reviewer djhm,
>
> We would like to thank you again for your efforts and time in providing constructive feedback and comments. We have carefully consider your advices/questions about baselines and the method design, and have provided as much refinement and experiments to address your concerns regarding the submission. Since there are now only **one day** remaining that we are allowed to further modify the submission. Would you mind checking the current **updated submission** and confirming whether we have solved your concerns. If there are any further questions or suggestions, we would like to conduct the experiments or refine the description to improve our submission.
>
> Sincerely
>
> The authors

---

> ### Author Response · Authors · 2022-11-18
> **Would you mind checking our response? Welcome for more discussions.**
>
> Dear Reviewer djhm,
>
> Thank you again for your time and efforts in reviewing our paper! As the end of Discussion Stage 1 is approaching and the window for paper revision is closing, here is a summary of our previous response and update:
>
> - Clarify that all our experiments were conducted under the framework of the state-of-the-art partial label learning methods to avoid misunderstandings. Add more comparisons with baselines that focus on imbalances under PLL (See the detailed response or Table 4 and Appendix E.6).
> - Clarify the contribution of our proposed method (See the detailed response), and add results and discussions about ablation study on other straightforward dynamic strategies (See Table 5).
>
> **We humbly expect you could check our responses with our updated version, and confirm whether our response has addressed your concerns. More discussions are always welcome. Please let us know if there are any further questions or suggestions that we could clarify or improve.**
>
> Sincerely,
>
> The authors

---

> ### Author Response · Authors · 2022-11-23
> **Looking forward to your responses or further suggestions/comments!**
>
> Dear Reviewer djhm,
>
> We have carefully considered and addressed your initial concerns regarding our paper. We are happy to discuss them with you in the openreview system if you feel that there still are some concerns/questions. We also welcome new suggestions/comments from you!
>
> Best regards,
>
> The authors of Paper3732

---

> ### Author Response · Authors · 2022-11-26
> **We anticipate your feedback!**
>
> Dear Reviewer djhm,
>
> The authors greatly appreciate your time and effort in reviewing this submission, and eagerly await your response.
>
> We have provided detailed responses to every one of your concerns/questions. Please help us to review our responses once again and kindly let us know whether they fully or partially address your concerns and if our explanations are in the right direction.
>
> Best,
>
> The authors of Paper3732

---

> ### Author Response · Authors · 2022-12-05
> **We are anticipating your post-rebuttal feedback!**
>
> Dear Reviewer djhm,
>
> We would like to sincerely thank you again for your time in reviewing our work!
>
> We understand you might be quite busy. However, as the discussion deadline is approaching, would you mind checking our response and confirming whether you have any further questions? Any further comments and discussions are welcomed!
>
> Best Regards,
>
> The authors of Paper3732

---

### Official Review · Reviewer_sVPa · 2022-11-02

**Confidence:** 2
**Clarity, Quality, Novelty And Reproducibility:** Please see above.
**Correctness:** 3
**Technical Novelty And Significance:** 2
**Empirical Novelty And Significance:** 2
**Recommendation:** 5

**Strength And Weaknesses:**

Strength:
- Proposed dynamic re-balancing technique is interesting.
- The problem formulation is theoretically interesting.

Weaknesses:
- The experimental results are not convincing. The main issue with this paper is that it does not compare its performance with the state of the art methods for longtailed learning. For example the following paper reports in excess of 52% accuracy on CIFAR-100 LT.

Jianggang Zhu, Zheng Wang, Jingjing Chen, Yi-Ping Phoebe Chen, Yu-Gang Jiang. Balanced Contrastive Learning for Long-Tailed Visual Recognition. CVPR 2022

- Another major problem is about the practical motivation for the proposed problem. Authors almost assume that LT+PLL is an important problem setting to learning with without giving any motivating examples. This is also reflected in the experimental set up on benchmark data sets where for the PLL setting the authors simply use uniform probability to generate partial label sets from CIFAR 100 LT. This might generate more partial labels for the tail classes.


- Finally, the paper is very difficult to read. This is because, the exposition in the paper assumes that the author is familiar with the literature on long-tailed learning and partial label learning. However many of these settings are not mainstream and the approaches should be introduced properly.

For example, prior to equation 2 the authors do not define p_uni(y) but refer to a previous paper. It is hard for a reader not familiar with the literature to understand the definition from the previous paper.


**Summary Of The Paper:**

This paper describes a new method for rebalancing of class probabilities for longtailed learning under the setting of partial label learning. No others followed the idea of logit adjustment, which factories P(y|x) as P(x|y) P(y). The second component is estimated on a per class basis in the training algorithm.

The authors show their experimental results on CIFAR-100 LT and CIFAR 10 LT data sets.

**Summary Of The Review:**

Overall the paper is well written and dwells on topical problems. However the current problem is poorly motivated and seems almost like a combination of two other problems. The proposed method is logical and simple. However the exposition of the proposed methods in my opinion needs to be more lucid. Finally while lot of experiments have been performed showing improvement over existing PLL techniques, the baselines for long tail learning have not been incorporated.

---

> ### Author Response · Authors · 2022-11-14
> **Response to Reviewer sVPa[2/2]**
>
> > **Q2**: Another major problem is about the practical motivation for the proposed problem. Authors almost assume that LT+PLL is an important problem setting to learning with without giving any motivating examples. This is also reflected in the experimental set up on benchmark data sets where for the PLL setting the authors simply use uniform probability to generate partial label sets from CIFAR 100 LT. This might generate more partial labels for the tail classes.
>
> **A2:** We appreciate the reviewer question about the motivating examples. Partial label learning origins from the real-world scenarios, where the annotation for each sample is an ambiguous set containing the groundtruth and the other confusing labels. This is common when we gather annotations of samples from news websites with several tags, videos with several characters of interest, or labels from multiple annotators in professional area like biology, medical area. However, the ideal assumption behind PLL is that the collected data is approximately uniformly distributed regarding classes, e.g., the benchmark datasets for a range of PLL explorations. However, a more practical setting regarding data should be imbalance, especially follow the natural long-tailed law, which should be considered in the area of PLL works if we consider to deploy the PLL methods into real-world applications like large-scale medical disease classification (some severe diseases only has few patient examples) and multi-person tracking in the basket videos (some non-stars only have a few close-up shots). This motivates us to consider the setting of LT + PLL.
>
> We will refine the introduction to emphasize the motivation from the perspective of real-world scenarios, and add a background section in the appendix to specially discuss some examples of LT-PLL.
>
> About the concern on the uniform probability to generate partial label sets, we do actually conduct the non-uniform probabilistic cases in Table 3, Table 9, and Table 11 in our revised draft. Besides, on the Pascal VOC dataset, candidate label sets are generated from co-occurring objects, which is also not the uniform probability.
>
> > **Q3**: Finally, the paper is very difficult to read. This is because, the exposition in the paper assumes that the author is familiar with the literature on long-tailed learning and partial label learning. However many of these settings are not mainstream and the approaches should be introduced properly. For example, prior to equation 2 the authors do not define p_uni(y) but refer to a previous paper. It is hard for a reader not familiar with the literature to understand the definition from the previous paper.
>
> **A3:** Thank you for the suggestion. We have completed the definition about $\mathbb{P}_{uni}(y)$ in that paragraph. Besides, we refined the description of the preliminary parts to be more clear, and completed more detailed technical introduction about partial label learning and long-tailed learning in the appendix to help the reviewers and the readers ease the reading.

---

> ### Author Response · Authors · 2022-11-14
> **Response to Reviewer sVPa[1/2]**
>
> Thank you for your time devoted to reviewing this paper and your constructive suggestions. Here are our detailed replies to your questions.
>
> > **Q1**: The experimental results are not convincing. The main issue with this paper is that it does not compare its performance with the state of the art methods for longtailed learning. For example the following paper reports in excess of 52\% accuracy on CIFAR-100 LT.
>
> **A1**: Thanks for recommending the work [1] and we will add it into the submission with the corresponding discussion. To alleviate the concern of the reviewer, we add two state-of-the-art long-tailed learning methods, BCL [1] and PaCO [2], into our experiments on basis of the state-of-the-art CORR in partial label learning. Similar to Oracle-LA, we apply the oracle class distributions for BCL and PaCO, denoted as Oracle-BCL and Oracle-PaCO. The results are summarized in the below tables.
>
> [**Table 1.** Comparison with different long-tailed learning methods on CIFAR-10-LT]
> | Imbalance ratio $\rho$ |  50   |  50   |  50   |  100  |  100  |  100  |
> | ---------------------- |:-----:|:-----:|:-----:|:-----:|:-----:|:-----:|
> | ambiguity $q$          |  0.3  |  0.5  |  0.7  |  0.3  |  0.5  |  0.7  |
> | CORR                   | 76.12 | 56.45 | 41.56 | 66.38 | 50.09 | 38.11 |
> | + Oracle-LA            | 36.27 | 17.61 | 12.77 | 29.97 | 15.80 | 11.75 |
> | + Oracle-LA-Posthoc    | 80.70 | 58.49 | 43.44 | 72.96 | 54.64 | 41.66 |
> | + Oracle-BCL           | 43.76 | 25.24 | 17.34 | 33.23 | 18.34 | 15.72 |
> | + Oracle-PaCO          | 42.45 | 26.34 | 16.23 | 34.21 | 18.23 | 14.23 |
> | + RECORDS              | **82.57** | **80.28** | **67.24** | **77.66** | **72.90** | **57.46** |
>
> [**Table 2.** Comparison with different long-tailed learning methods on CIFAR-100-LT]
> | Imbalance ratio $\rho$ |     50     |     50     |     50     |     100    |     100    |     100    |
> |------------------------|:----------:|:----------:|:----------:|:----------:|:----------:|:----------:|
> | ambiguity $q$          | 0.03       | 0.05       | 0.07       | 0.03       | 0.05       | 0.07       |
> | CORR                   | 42.29      | 38.03      | 36.59      | 38.39      | 34.09      | 31.05      |
> | + Oracle-LA            | 22.56      | 5.59       | 3.12       | 11.37      | 3.32       | 1.98       |
> | + Oracle-LA-Posthoc    | 46.94      | 40.76      | 39.07      | 41.49      | 36.79      | 33.32      |
> | + Oracle-BCL           | 27.23      | 10.34      | 8.34       | 14.34      | 5.82       | 4.10       |
> | + Oracle-PaCO          | 27.37      | 11.23      | 7.23       | 15.82      | 5.62       | 4.72       |
> | + RECORDS| **48.06** | **45.56** | **42.51** | **42.25** | **40.59** | **38.65** |
>
> From the tables, we can find: 1) The performance under Oracle-BCL and Oracle-PaCO is better than that under Oracle-LA, meaning that a better logit adjustment method does help improve the model; 2) However, the performance of Oracle-BCL and Oracle-PaCO still fails to outperform our method with a large gap. The reason behind the failure attributes to our analysis in Figure 1, where existing long-tailed learning methods, especially logit adjustment and its variants, are lack of consideration about the training dynamic of PLL, and thus is not benign to label disambiguation in PLL. This motivates the proposal of our RECORDS. We will include more variants of logit-adjustment methods into the discussion and experiments, and add more complete results into the appendix.
>
> [1] Zhu, J., Wang, Z., Chen, J., Chen, Y. P. P., \& Jiang, Y. G. Balanced Contrastive Learning for Long-Tailed Visual Recognition. In Proceedings of the IEEE/CVF Conference on Computer Vision and Pattern Recognition, 2022.
>
> [2] Cui, J., Zhong, Z., Liu, S., Yu, B., \& Jia, J. Parametric contrastive learning. In Proceedings of the IEEE/CVF international conference on computer vision, 2021.

---

> ### Author Response · Authors · 2022-11-17
> **Kind Reminder to Respond Our Rebuttal**
>
> Dear Reviewer sVPa,
>
> We would like to thank you again for your efforts and time in providing constructive feedback and comments. We have carefully considered your suggestions/questions about long-tailed learning baselines, motivating examples of LT-PLL, and writing, and have provided as much refinement, experiments and supplemental materials to address your concerns regarding the submission. Since there are now only **one day** remaining that we are allowed to further modify the submission. Would you mind checking the current **updated submission** and confirming whether we have solved your concerns. If there are any further questions or suggestions, we would like to conduct the experiments or refine the description to improve our submission.
>
> Sincerely
>
> The authors

---

> ### Author Response · Authors · 2022-11-18
> **Would you mind checking our response? Welcome for more discussions.**
>
> Dear Reviewer sVPa,
>
> Thank you again for your time and efforts in reviewing our paper! As the end of Discussion Stage 1 is approaching and the window for paper revision is closing, here is a summary of our previous response and update:
>
> - Add results and discussions about more long-tailed learning baselines (See Table 4 and Appendix E.6).
> - Refine the introduction to emphasize the practical motivation of LT-PLL, and add a background section in the appendix to specially discuss some examples of LT-PLL (See Section 1 and Appendix A). Clarify that we have constructed experiments under non-uniform cases to avoid misunderstandings.
> - Enrich the description in the preliminary parts and complete more detailed introduction of two research lines to ease the understanding of the technical parts (See Section 3 and Appendix B).
>
> **We humbly expect you could check our responses with our updated version, and confirm whether our response has addressed your concerns. More discussions are always welcome. Please let us know if there are any further questions or suggestions that we could clarify or improve.**
>
> Sincerely,
>
> The authors

---

> ### Author Response · Authors · 2022-11-23
> **Looking forward to your responses or further suggestions/comments!**
>
> Dear Reviewer sVPa,
>
> We have carefully considered and addressed your initial concerns regarding our paper. We are happy to discuss them with you in the openreview system if you feel that there still are some concerns/questions. We also welcome new suggestions/comments from you!
>
> Best regards,
>
> The authors of Paper3732

---

> ### Author Response · Authors · 2022-11-26
> **We anticipate your feedback!**
>
> Dear Reviewer sVPa,
>
> The authors greatly appreciate your time and effort in reviewing this submission, and eagerly await your response.
>
> We have provided detailed responses to every one of your concerns/questions. Please help us to review our responses once again and kindly let us know whether they fully or partially address your concerns and if our explanations are in the right direction.
>
> Best,
>
> The authors of Paper3732

---

> ### Author Response · Authors · 2022-12-05
> **We are anticipating your post-rebuttal feedback!**
>
> Dear Reviewer sVPa,
>
> We would like to sincerely thank you again for your time in reviewing our work!
>
> We understand you might be quite busy. However, as the discussion deadline is approaching, would you mind checking our response and confirming whether you have any further questions? Any further comments and discussions are welcomed!
>
> Best Regards,
>
> The authors of Paper3732

---

### Author Response · Authors · 2022-11-15
**General Response by Authors**

We would like to thank all the reviewers for their thoughtful suggestions on our paper, and appreciate that the reviewers have some positive impressions of our work, including: (1) a well-motivated problem (djhm, dMZw, MogU); (2) a good demonstration of the problems with existing methods (djhm, dMZw); (3) an interesting and reasonable solution (sVPa, dMZw, MogU); (4) well-writen and clearly explained (djhm, dMZw, MogU); (5) being effective and successful supported by experimental results (dMZw, MogU).

According to the advice, we have carefully revised our draft with the proofreading to correct some typos and mistakes, and complete a range of experiments to address the concerns of the reviewers. In the following, we provide a summary of our updates, and for detailed responses, please refer to the feedback of each comment/question point-by-point and the new empirical evaluations.

1. We carefully refine the introduction about the motivation and the proper claim of the contribution to avoid the potential misunderstanding on novelty (See Section 1), and specially add a background section to discuss some practical motivating examples of LT-PLL (See Appendix A).
2. We enrich the description in the preliminary parts (See Section 3) and complete more detailed introduction of two research lines to ease the understanding of the technical parts (See Appendix B), add necessary citations and discussion for Eq.(4) about its rationality and intuition, and add more description near our Theorem 1 to admit previous theoretical contribution and our emphasis in this study (See Section 4 and Appendix D).
3. We comprehensively compare with four new baselines (BCL, PaCO, LDD, and SFN) to demonstrate the effectiveness of our method (See Table 4 and Appendix E.6), and conduct more ablation study by combining with other straightforward dynamic strategies to support the significance of our method (See Table 5).

The above updates in the revised draft (including the regular pages and the appendix) are highlighted in blue color.

We appreciate all reviewers’ time again and kindly invite the reviewers to check our revised draft for your concerns. We are looking forward to your reply!

---

### Public Comment · ~Haobo_Wang1 · 2023-02-08
**Nice work! Recommend our concurrent work on imbalanced partial label learning.**

Hi,

Nice work and congratulations! Our recent NeurIPS'22 paper SoLar (https://openreview.net/forum?id=wUUutywJY6) also studied imbalanced partial label learning based on optimal transport. Would you please consider citing our work?

Best
Haobo

---

> ### Author Response · Authors · 2023-02-08
> **Thank you for recommendation**
>
> Thanks, Haobo. We will cite your nice work and add the corresponding discussion. Actually, we meant to compare early but have no chance to access to your arxiv paper in time. Our camera ready version will include the discussion and comparison with your excellent method (specially in the appendix). We will also release our code and please refer to the incoming camera ready version with the code url.
>
> Best,
>
> Feng Hong

---

### Decision · Program_Chairs · 2023-01-20

**Decision:**

Accept: poster

**Justification For Why Not Higher Score:**

Experimental results are only on a small number of artificially manipulated standard datasets. There is no indication of value on a specific real-world application.

As the paper says, it "does not involve any human subjects, ... data set releases, potentially harmful insights, methodologies and applications, potential conflicts of interest and sponsorship, discrimination/bias/fairness concerns, privacy and security issues, legal compliance, and research integrity issues." The real-world importance is hypothetical, and the mathematical contribution is slight.

**Justification For Why Not Lower Score:**

The core idea of this paper is in Sections 3.3 and 4. The new method is based on a simple but effective and guaranteed previous method for dealing with imbalance, namely logit adjustment (3.3). The authors identify why this does not apply directly to partial label learning, and design a well-motivated extension (4).

**Metareview: Summary, Strengths And Weaknesses:**

(a) The paper provides a new method to overcome class imbalance in learning from data where among multiple training labels, many may be incorrect (PLL).

(b) Strengths: The work is thorough and the authors responded to the reviewers carefully.

(c) The work is incremental. It is a typical ML paper in the sense that it proposes a new algorithm that is a combination of known components, is heuristic, and is somewhat complicated. The experiments are merely on known datasets, with no new real-world application or insight.

**Note From Pc:**

if the above contains the word "oral" or "spotlight" please see: "oral" presentation means -> notable-top-5% and "spotlight" means -> notable-top-25%. As stated in our emails, we are disassociating presentation type from AC recommendations

**Summary Of Ac-Reviewer Meeting:**

No meeting.